# Feature-based data assimilation in geophysics

Matthias Morzfeld[1], Jesse Adams[1], Spencer Lunderman[1], and Rafael Orozco[1]

[1]Department of Mathematics, University of Arizona, 617 N. Santa Rita Ave., P.O. Box 210089, Tucson, Arizona 85721, USA

*Correspondence to:* Matthias Morzfeld, mmo@math.arizona.edu

**Abstract.** Many applications in science require that computational models and data be combined. In a Bayesian framework, this is usually done by defining likelihoods based on the mismatch of model outputs and data. However, matching model outputs and data in this way can be unnecessary or impossible. For example, using large amounts of steady state data is unnecessary because these data are redundant. It is numerically difficult to assimilate data in chaotic systems. It is often impossible to assimilate data of a complex system into a low-dimensional model. As a specific example, consider a low-dimensional stochastic model for the dipole of Earth's geomagnetic field, while other field components are neglected in the model. The above issues can be addressed by selecting features of the data, and defining likelihoods based on the features, rather than by the usual mismatch of model output and data. Our goal is to contribute to a fundamental understanding of such a feature-based approach that allows us to assimilate selected aspects of data into models. We also explain how the feature-based approach can be interpreted as a method for reducing an effective dimension and derive new noise models, based on perturbed observations, that lead to computationally efficient solutions. Numerical implementations of our ideas are illustrated in four examples.

## 1 Introduction

The basic idea of data assimilation is to update a computational model with information from sparse and noisy data so that the updated model can be used for predictions. Data assimilation is at the core of computational geophysics, e.g., in numerical weather prediction (Bauer et al., 2015), oceanography (Bocquet et al., 2010) and geomagnetism (Fournier et al., 2010). Data assimilation is also used in engineering applications, e.g., in robotics (Thrun et al., 2005) and reservoir modeling (Oliver et al., 2008). We use the term "data assimilation" broadly, but focus on parameter estimation problems where one attempts to find model parameters such that the output of the model matches data. This is achieved by defining a posterior distribution that describes the probabilities of model parameters conditioned on the data.

The posterior distribution is proportional to the product of a prior distribution and a likelihood. The likelihood connects the model and its parameters to the data and is often based on the mismatch of model output and data. A typical example is the squared two-norm of the difference of model output and data. However, estimating model parameters based on such a direct mismatch of model outputs and data may not be required or feasible. It is not required, for example, if the data are intrinsically low-dimensional, or if the data are redundant (we discuss a specific example in section 4.1). Examples of situations where data assimilation is infeasible can be classified into two groups. First, the model may be lower-dimensional than the data. This situation occurs when selected aspects of a complex system are represented by a low-dimensional model. Examples include

low-dimensional modeling of Earth's dipole for time-scales of millions of years as discussed, e.g., in Gissinger (2012); Petrelis et al. (2009); Buffett et al. (2013); Buffett and Matsui (2015). These simplified models cannot represent all aspects of Earth's magnetic field and, hence, using observations of Earth's magnetic field for parameter or state estimation with these models is not possible. We will elaborate on this example in section 4.3. Another example of low-dimensional models for complex

processes are the simplified delay-differential equations, used by Koren and Feingold (2011); Feingold and Koren (2013); Koren et al. (2017), to model behaviors of cloud systems over warm oceans. In both cases, model outputs cannot directly match data, because the low-dimensional model was not designed to capture all aspects of a complex system (clouds or Earth's magnetic field). Second, matching model outputs to data directly becomes numerically impossible if one considers chaotic models over long time-scales. We will discuss this case in detail in section 4.4.

The above issues of data assimilation can be addressed by adapting ideas from machine learning to data assimilation. Machine learning algorithms expand the data into a suitable basis of "feature vectors" (Murphy, 2012; Bishop, 2006; Rasmussen and Williams, 2006). A feature can be thought of as a low-dimensional representation of the data, e.g., a principal component analysis (PCA) (Jolliffe, 2014), a Gaussian process model (Rasmussen and Williams, 2006), or a Gaussian mixture model (McLachlan and Peel, 2000). Features are either constructed a priori, or learned from data. The same ideas carry over to

data assimilation. One can extract low-dimensional features from the data and use the model to reproduce these features. A feature-based likelihood can be constructed to measure the mismatch of the observed features and the features produced by the model. The feature-based likelihood and a prior distribution define a feature-based posterior distribution, which describes the probability of model parameters conditioned on the features. We discuss only features that are constructed a priori and using physical insight into the problem. Learning features "automatically" from data is the subject of future work.

As a specific example, consider a viscously damped harmonic oscillator, defined by damping and stiffness coefficients (we assume we know its mass). An experiment may be to pull on the mass and then to release it and to measure the displacement of the mass from equilibrium as a function of time. These data can be compressed into features in various ways. For example, a feature could be the statement that "the system exhibits oscillations". Based on this feature, one can infer that the damping coefficient is less than one. Other features may be the decay rate or observed oscillation frequency. One can compute the

damping and stiffness coefficients using classical formulas, if these quantities were known exactly. The idea of feature-based data assimilation is to make such inferences in view of uncertainties associated with the features.

Another example is Lagrangian data assimilation for fluid flow, where the data are trajectories of tracers and where a natural candidate for a feature is a coherent structure (Maclean et al., 2017). The coherent structure can be used to formulate a likelihood, which in turn defines a posterior distribution that describes the probability of model parameters given the observed

coherent structure, but without direct appeal to tracer trajectories. More generally, consider a chaotic system observed over long time scales, e.g., several $e$-folding times of the system. Due to the chaotic behavior, changes in the numerical differential equation solver may change likelihoods based on model-output/data mismatch, even if the parameters and data remain unchanged. The feature-based approach can be useful here, as shown by Hakkarainen et al. (2012), who use likelihoods based on particle filter runs to average out uncertainties from differential equation solvers. Haario et al. (2015) use correlation vectors and

summary statistics, which are "features" in our terminology, to identify parameters of chaotic systems such as the Lorenz'63 (Lorenz, 1963) and Lorenz'95 (Lorenz, 1995) equations.

Our goal is to contribute to a fundamental understanding of the feature-based approach to data assimilation and to extend the numerical framework for solving feature-based data assimilation problems. We also discuss the conditions under which the feature-based approach is appropriate. In this context, we distinguish two problem classes. First, the compression of the data into a feature may lead to no or little loss of information, in which case the feature-based problem and the "original" problem, as well as their solutions, are similar. Specific examples are intrinsically low-dimensional data or redundant (steady state) data. Second, the features extracted from the data may be designed to deliberately neglect information in the data. This second case is more interesting because we can assimilate selected aspects of data into low-dimensional models for complex systems and we can formulate feature-based problems that lead to useful parameter estimates for chaotic systems, for which a direct approach is computationally expensive or numerically infeasible. We give interpretations of these ideas in terms of effective dimensions of data assimilation problems (Chorin and Morzfeld, 2013; Agapiou et al., 2016) and interpret the feature-based approach as a method for reducing the effective dimension. Our discussion as well as our numerical examples suggest that the feature-based approach is comparable to a direct approach when the data can be compressed without loss of information and that computational efficiency is gained only when the features truly reduce the dimension of the data, i.e., if some aspects of the data are indeed neglected.

Nonetheless, the feature-based likelihood can be cumbersome to evaluate. The reason is that an evaluation of a feature-based likelihood may involve repeated solution of stochastic equations, followed by a compression of a large amount of simulation data into features and it is unclear how to assess the error statistics of the features. In fact, the inaccessible likelihood prevents application of the typical numerical methods for data assimilation, e.g., Monte Carlo sampling or optimization. We suggest to overcome this difficulty by adapting ideas from stochastic ensemble Kalman filters (Evensen, 2006) and to derive noise models directly for the features using "perturbed observations". Such noise models lead to feature-based likelihoods which are easy to evaluate, so that Monte Carlo methods can be used for the solution of feature-based data assimilation problems. Another numerical difficulty is that the feature-based likelihood can be noisy, e.g., if it is based on averages computed by Monte Carlo simulations. In such cases, we suggest to apply numerical optimization to obtain maximum a posteriori estimates, rather than Monte Carlo methods, because optimization is more robust to noise.

Details of the numerical solution of feature-based data assimilation problems are discussed in the context of four examples, two of which involve "real" data. Each example represents its own challenges and we suggest appropriate numerical techniques, including Markov Chain Monte Carlo (MCMC, (Kalos and Whitlock, 1986)), direct sampling (see, e.g., Chorin and Hald (2013); Owen (2013)) and global Bayesian optimization (see, e.g., Frazier and Wang (2016)). The variety of applications and the variety of numerical methods we can use to solve these problems indicate the flexibility and usefulness of the feature-based approach.

Ideas related to ours were recently discussed by Rosenthal et al. (2017) in the context of data assimilation problems in which certain geometric features need to be preserved. This situation occurs, e.g., when estimating wave characteristics, or tracking large scale structures such as storm systems. Data assimilation typically does not preserve geometric features but Rosenthal

et al. (2017) use kinematically constrained transformations to preserve geometric features within an ensemble Kalman filtering framework. The techniques discussed by Rosenthal et al. (2017) are related to the feature-based data assimilation we describe here, but they different at its core and its goals: Rosenthal et al. (2017) are concerned about *preserving* features during data assimilation while we wish to *estimate* model parameters from features. We further emphasize that a feature-based approach may also be useful when high-fidelity models, such as coupled ocean-hurricane models, are used. In this case, one may need to reduce the dimension of some of the data and assimilate only some features into the high-dimensional model. This is discussed in Falkovich et al. (2005); Yablonsky and Ginis (2008). Here, we focus on problems in which the data are high-dimensional, but the model is low-dimensional.

## 2 Background

We briefly review the typical data assimilation problem formulation and several methods for its numerical solution. The descriptions of the numerical techniques may not be sufficient to fully comprehend the advantages or disadvantages of each method, but these are explained in the references we cite.

### 2.1 Data assimilation problem formulation

Suppose you have a mathematical/computational model $\mathcal{M}$ that maps input parameters $\theta$ to outputs $y$, i.e., $y = \mathcal{M}(\theta)$ where $\theta$ and $y$ are $n$- and $k$-dimensional real vectors. The parameters $\theta$ may be initial or boundary conditions of a partial differential equation, diffusion coefficients in elliptic equations, or growth rates in ecological models. The outputs $y$ can be compared to data $z$, obtained by observing the physical process under study. For example, if $\mathcal{M}$ is an atmospheric model, $z$ may represent temperature measurements at $k$ different locations. It is common to assume that

$$z = \mathcal{M}(\theta) + \varepsilon, \tag{1}$$

where $\varepsilon$ is a random variable with known probability density function (pdf) $p_\varepsilon(\cdot)$ that describes errors/mismatch between model and data. The above equation defines a $k$-dimensional "likelihood", $l(z|\theta) = p_\varepsilon(z - \mathcal{M}(\theta)|\theta)$, that describes the probability of the data for a given set of parameters.

In addition to equation (1), one may have prior information about the model parameters, e.g., one may know that some parameters are non-negative. Such prior information can be represented by a prior distribution $p_0(\theta)$. By Bayes' rule, the prior and likelihood define the posterior distribution

$$p(\theta|z) \propto p_0(\theta)\, l(z|\theta). \tag{2}$$

The posterior distribution combines information from model and data and defines parameters $\theta$ that lead to model outputs that are "compatible" with the data. Here compatible means that model outputs are likely to be within the assumed errors $\varepsilon$.

Data assimilation problems of this kind appear in science and engineering, e.g., in numerical weather prediction, oceanography and geomagnetism (Bocquet et al., 2010; van Leeuwen, 2009; Fournier et al., 2010), as well as in global seismic inversion

(Bui-Thanh et al., 2013), reservoir modeling/subsurface flow (Oliver et al., 2008), target tracking (Doucet et al., 2001) and robotics (Thrun et al., 2005; Morzfeld, 2015). The term "data assimilation" is common in geophysics, but in various applications and disciplines, different names are used, including parameter estimation, Bayesian inverse problems, history matching and particle filtering.

## 2.2 Numerical methods for data assimilation

Computational methods for data assimilation can be divided into three groups. The first group is based on the Kalman filter (Kalman, 1960; Kalman and Bucy, 1961) and includes e.g., the ensemble Kalman filter (Evensen, 2006). Kalman filters are particularly useful when data are assimilated sequentially, as is the case in numerical weather prediction. The second group consists of optimization algorithms, called "variational methods" in this context (Talagrand and Courtier, 1987). The third group are Monte Carlo sampling methods, including particle filters/direct sampling (Owen, 2013; Doucet et al., 2001; Atkins et al., 2013; Morzfeld et al., 2015) and Markov Chain Monte Carlo (MCMC) (Mackay, 1998; Kalos and Whitlock, 1986). We will use variational methods, MCMC and direct sampling for numerical solution of feature-based data assimilation problems and we briefly review these techniques here. We do not use Kalman filter techniques because the problems we discuss do not have an explicit time-dependence.

In variational data assimilation one finds the parameter set $\theta^*$ that maximizes the posterior probability, which is also called the posterior mode. One can find the posterior mode by minimizing the negative logarithm of the posterior distribution

$$F(\theta) = -\log\left(p_0(\theta)\, l(z|\theta)\right). \tag{3}$$

The optimization is done numerically and one can use, e.g., Gauss-Newton algorithms. In some of the numerical examples below, we need to optimize functions $F(\theta)$ that are computationally expensive to evaluate and noisy, i.e., $F(\theta)$ is a random variable with unknown distribution. The source of noise in the function $F(\theta)$ is caused by numerically approximating the feature. Suppose, e.g., that the feature is an expected value and in the numerical implementation this expected value is approximated by Monte Carlo. The Monte Carlo approximation, however, depends on the number of samples used and if this number is small (finite), the approximation is noisy, i.e., the Monte Carlo average for *the same* set of parameters $\theta$, but with two different seeds in the random number generator, can lead to two different values for $F(\theta)$. In such cases, one can use a derivative free optimization method such as global Bayesian optimization (GBO), see, e.g., Frazier and Wang (2016). The basic idea is to model the function $F(\theta)$ by a Gaussian process (GP) and then update the GP model based on a small number of function evaluations. The points where the function is evaluated are chosen based on an expected improvement (EI) criterion, which takes into account where the function is unknown or known. The GP model for the function $F(\theta)$ is then updated based on the function evaluations at the points suggested by EI. One can iterate this procedure and when the iteration is finished, e.g., because a maximum number of function evaluations is reached, one can use the optimizer of the mean of the GP model to approximate the optimizer of the (random) function $F(\theta)$.

In Markov Chain Monte Carlo (MCMC), a Markov chain is generated by drawing a new sample $\theta'$ given a previous sample $\theta^{j-1}$, using a proposal distribution $q(\theta'|\theta^{j-1})$. The proposed sample $\theta'$ is accepted as $\theta^j$ or rejected based on the values of

the posterior distribution of the new and previous samples, see, e.g., Mackay (1998); Kalos and Whitlock (1986). Averages over the samples converge to expected values with respect to the posterior distribution as the number of samples goes to infinity. However, since $\theta^j$ depends on $\theta^{j-1}$, the samples are not independent and one may wonder how many effectively uncorrelated samples one has obtained. This number can be estimated by dividing the number of samples by the integrated auto-correlation time (IACT) (Mackay, 1998; Kalos and Whitlock, 1986). Thus, one wants to pick a proposal distribution that reduces IACT. The various MCMC algorithms in the literature differ in how the proposal distribution is constructed. In the numerical examples below, we use the MATLAB implementation of the affine invariant ensemble sampler (Goodman and Weare, 2010), as described by Grinsted (2017), and we use the numerical methods described in Wolff (2004) to compute IACT.

In direct sampling (sometimes called importance sampling) one generates independent samples using a proposal density $q$ and attaches to each sample a weight:

$$\theta^j \sim q(\theta^j), \quad w^j \propto \frac{p_0(\theta^j)\, l(z|\theta^j)}{q(\theta_j)}.$$

Weighted averages of the samples converge to expectations with respect to the posterior distribution as the number of samples goes to infinity. While the samples are independent, they are not all equally weighted and one may wonder how many "effectively unweighted" samples one has. For an ensemble of size $N_e$, the effective number of samples can be estimated as (Doucet et al., 2001; Arulampalam et al., 2002)

$$N_{\text{eff}} = N_e/\rho, \quad \rho = \frac{E(w^2)}{E(w)^2}. \tag{4}$$

For a practical algorithm, we thus chose an importance function $q$ such that $\rho$ is near one. There are several strategies for constructing such proposal distributions and in the numerical illustrations below we use "implicit sampling" (Chorin and Tu, 2009; Morzfeld et al., 2015; Chorin et al., 2015) and construct the proposal distribution to be a Gaussian whose mean is the posterior mode $\theta^*$ and whose covariance is the Hessian of $F$ in (3), evaluated at $\theta^*$ (see also Owen (2013)).

## 3 Feature-based data assimilation

The basic idea of feature-based data assimilation is to replace the data assimilation problem defined by a prior $p_0(\theta)$ and the likelihood in (1) by another problem that uses only selected features of the data $z$. We assume that the prior is appropriate and focus on constructing new likelihoods. Let $\mathcal{F}(\cdot)$ be an $m$-dimensional vector function that takes a $k$-dimensional data set into a $m$-dimensional feature. One can apply the feature-extraction to equation (1) and obtain

$$f = \mathcal{F}(\mathcal{M}(\theta) + \varepsilon), \tag{5}$$

where $f = \mathcal{F}(z)$, is the feature extracted from the data $z$. Equation (5) can be used to define a feature-based likelihood $l_{\mathcal{F}}(f|\theta)$, which in turn defines a feature-based posterior distribution, $p_{\mathcal{F}}(\theta|f) \propto p_0(\theta)\, l_{\mathcal{F}}(f|\theta)$. The feature-based posterior distribution describes the probabilities of model parameters conditioned on the feature $f$ and can be used to make inferences about the parameters $\theta$.

### 3.1 Noise modeling

Evaluating the feature-based posterior distribution is difficult because evaluating the feature-based likelihood is cumbersome. Even under simplifying assumptions of additive Gaussian noise in equation (1), the likelihood $l_{\mathcal{F}}(f|\theta)$, defined by equation (5), is generally not known. The reason is that the feature function $\mathcal{F}$ makes the distribution of $\mathcal{F}(\mathcal{M}(\theta) + \varepsilon)$ non-Gaussian even

if $\varepsilon$ is Gaussian. Thus, the feature-based likelihood is Gaussian only if $\mathcal{F}$ is linear and if $\varepsilon$ is Gaussian. Numerical methods for data assimilation typically require that the posterior distribution be known up to a multiplicative constant. This is generally not the case when a feature-based likelihood is used. Thus, variational methods, MCMC or direct sampling are not directly applicable to solve feature-based data assimilation problems defined by (5). More advanced techniques, such as approximate Bayesian computation (ABC) (Marin et al., 2012), however, can be used because these are designed for problems with unknown

likelihood (see Maclean et al. (2017)).

Difficulties with evaluating the feature-based likelihood arise because we assume that equation (1) is accurate and we require that the feature-based likelihood follows directly from it. However, in many situations the assumptions about the noise $\varepsilon$ in equation (1) are "ad hoc", or for mathematical and computational convenience. There is often no physical reason why the noise should be additive or Gaussian, yet these assumptions have become standard in many data assimilation applications. This leads

to the question: *why not "invent" a suitable and convenient noise model for the feature*?

We explore this idea and consider an additive Gaussian noise model for the feature. This amounts to replacing equation (5) by

$$f = \mathcal{M}_{\mathcal{F}}(\theta) + \eta, \tag{6}$$

where $\mathcal{M}_{\mathcal{F}} = \mathcal{F} \circ \mathcal{M}$, is the composition of the model $\mathcal{M}$ and feature extraction $\mathcal{F}$ and $\eta$ is a random variable that represents

uncertainty in the feature and which we need to define (see below). For a given $\eta$, the feature-based likelihood defined by (6), $l_{\mathcal{F}}(f|\theta) = p_{\eta}(f - \mathcal{M}_{\mathcal{F}}(\theta)|\theta)$, is now straightforward to evaluate (up to a multiplicative constant) because the distribution of $\eta$ is known/chosen. The feature-based likelihood based on (6) results in the feature-based posterior distribution

$$p_{\mathcal{F}}(\theta|f) \propto p_0(\theta)\, l_{\mathcal{F}}(f|\theta), \tag{7}$$

where $p_0(\theta)$ is the prior distribution, which is not affected by defining or using features. The usual numerical tools, e.g.,

MCMC, direct sampling, or variational methods, are applicable to the feature-based posterior distribution (7).

Our simplified approach requires that one defines the distribution of the errors $\eta$, similar to how one must specify the distribution of $\varepsilon$ in equation (1). We suggest to use a Gaussian distribution with mean zero for $\eta$. The covariance that defines the Gaussian can be obtained by borrowing ideas from the ensemble Kalman filter (EnKF). In a "perturbed observation" implementation of the EnKF, the analysis ensemble is formed by using artificial perturbations of the data (Evensen, 2006). We

suggest to use a similar approach here. Assuming that $\varepsilon$ in equation (1) is Gaussian with mean $0$ and covariance matrix $R$, we generate perturbed data by: $z^j \sim \mathcal{N}(z, R), j = 1, \ldots, N_z$. Each perturbed data leads to a perturbed feature $f^j \sim \mathcal{F}(z^j)$ and we

compute the covariance

$$R_f = \frac{1}{N_z - 1} \sum_{j=1}^{N_z} (f^j - f)(f^j - f)^T.$$

We then use $\eta \sim \mathcal{N}(0, R_f)$ as our noise model for the feature-based problem in equation (6).

Note that the rank of the covariance $R_f$ is $\min\{\dim(f), N_z - 1\}$. For high-dimensional features, the rank of $R_f$ may therefore be limited by the number of perturbed observations and features we generate, and this number depends on the computational requirements of the feature extraction. We assume that $N_z$ is larger than the dimension of the feature, which is the case if either the computations to extract the features are straightforward, or if the feature is low-dimensional.

One may also question why $\eta$ should be Gaussian. In the same vein, one may wonder why $\varepsilon$ in equation (1) should be Gaussian, which is routinely assumed. We do not claim that we have answers to such questions, but we speculate that if the feature does indeed constrain some parameters, then assuming a unimodal likelihood is appropriate and, in this case, a Gaussian approximation is also appropriate.

## 3.2 Feature selection

Feature-based data assimilation requires that one defines and selects relevant features. In principle, much of the machine learning technology can be applied to extract generic features from data. For example, one can define $\mathcal{F}$ by the PCA, or singular value decomposition, of the data and then neglect small singular values and associated singular vectors. As a specific example, suppose that the data are measurements of a time series of $M$ data points of an $n$-dimensional system. In this case, the function $\mathcal{M}_{\mathcal{F}}$ consists of the steps (*i*) simulate the model; and (*ii*) compute the SVD of the $n \times M$ matrix containing the data. The feature $f$ in (6) may then be the first $l$ largest singular values and associated right and left singular vectors (see the example in section 4.2 for more detail). In practice, relevant features may often present themselves. For example, in Lagrangian data assimilation, coherent structures (and their SVDs) are a natural candidate, as explained by Maclean et al. (2017). In section 4 we present several examples of "intuitive" features, constructed using physical insight, and discuss what numerical methods to use in the various situations.

The choice of the feature suggests the numerical methods for the solution of the feature-based problem. One issue here is that, even with our simplifying assumption of additive (Gaussian) noise in the feature, evaluating a feature-based likelihood can be noisy. This happens in particular when the feature is defined in terms of averages over solutions of stochastic or chaotic equations. Due to limited computational budgets, such averages are computed using a small sample size. Thus, sampling error is large and evaluation of a feature-based likelihood is noisy, i.e., evaluations of the feature-based likelihood, even for the same set of parameters $\theta$ and feature-data $f$, may lead to different results, depending on the state of the random number generator. This additional uncertainty makes it difficult to solve some feature-based problems numerically by Monte Carlo. However, one can construct a numerical framework for computing maximum a posteriori estimates using derivative free optimization methods that are robust to noise, e.g., global Bayesian optimization (Frazier and Wang, 2016). We will specify these ideas in the context of a numerical example with the Kuramoto-Sivashinsky equation in Section 4.4.

## 3.3 When is a feature-based approach useful?

A natural question is: *under what conditions should I consider a feature-based approach*? There are three scenarios which we discuss separately before we make connections between the three scenarios using the concept of "effective dimension".

### 3.3.1 Case (i): data compression *without* information loss

It may be possible that data can be compressed into features without significant loss of information, for example, if observations are collected while a system is in steady state. Steady state data are redundant, make negligible contributions to the likelihood and posterior distributions and, therefore, can be neglected. This suggests that features can be based on truncated data and that the resulting parameter estimates and posterior distributions are almost identical to the estimates and posterior distributions based on *all* the data. We provide a detailed numerical example to illustrate this case in section 4.1. Similarly, suppose the feature is based on the PCA of the data, e.g., only the first $l$ singular values and associated singular vectors are used. If the neglected singular values are indeed small, then the data assimilation problem defined by the feature, i.e., the truncated PCA, and the data assimilation problem defined by all of the data, i.e., the full set of singular values and singular vectors, are essentially the same. We discuss this in more detail and with the help of a numerical example in section 4.2.

### 3.3.2 Case (ii): data compression *with* information loss

In some applications a posterior distribution defined by all of the data may not be practical or computable. An example is estimation of initial conditions and other parameters based on (noisy) observations of a chaotic system over long time-scales. In a "direct" approach one tries to estimate initial conditions that lead to trajectories that are near the observations at all times. Due to the sensitivity to initial conditions a point-wise match of model-output and data is numerically difficult to achieve. In a feature-based approach one does not insist on a point-by-point match of model-output and data, i.e., the feature-based approach *simplifies* the problem by neglecting several important aspects of the data during the feature-extraction (e.g., the time-ordering of the data points). As a specific example consider estimation of model parameters that lead to trajectories with similar characteristics to the observed trajectories. If only some characteristics of the trajectories are of interest, then the initial conditions need never be estimated. Using features thus avoids the main difficulty of this problem (extreme sensitivity to small perturbations) provided one can design and extract features that are robust across the attractor. Several examples have already been reported where this is indeed the case, see Hakkarainen et al. (2012); Haario et al. (2015); Maclean et al. (2017). We will provide another example and additional explanations, in particular about feature selection and numerical issues, in section 4.4. It is important to realize that the *solution* of the feature-based problem is different from the solution of the (unsolvable) original problem because several important aspects of the data have been neglected. In particular, we emphasize that the solution of the feature-based problem yields parameters that lead to trajectories with similarities with the data, as defined by the feature. The solution of the (possibly infeasible) original problem yields model parameters that lead to trajectories that exhibit a good point-by-point match with the data.

### 3.3.3 Case (iii): models and data at different scales

The feature-based approach is essential for problems for which the numerical model and the data are characterized by different scales (spatial, temporal or both). Features can be designed to filter out fine scales that may be present in the data, but which are not represented by the numerical model. This is particularly important when a low-dimensional model is used to represent certain aspects of a complex system. Specific examples of low-dimensional models for complex processes can be found in the modeling of clouds or the geomagnetic field (Gissinger, 2012; Petrelis et al., 2009; Buffett et al., 2013; Buffett and Matsui, 2015; Koren and Feingold, 2011; Feingold and Koren, 2013). Methods that evaluate the skill of these models in view of data are missing and the feature-based approach may be useful in this context. We discuss this case in more detail and with the help of a numerical example in section 4.3.

### 3.3.4 Reduction of effective dimension

Cases (i) and (ii) can be understood more formally using the concept of an "effective dimension". The basic idea is that a high-dimensional data assimilation problem is more difficult than a low-dimensional problem. However, it is not only the number of parameters that defines dimension in this context, but rather a combination of the number of parameters, the assumed distributions of errors and prior probability, as well as the number of data points (Chorin and Morzfeld, 2013; Agapiou et al., 2016). An effective dimension describes this difficulty of a data assimilation problem, taking into account all of the above, and is focused on the computational requirements of numerical methods (Monte Carlo) to solve a given problem: a low effective dimension means the computations required to solve the problem are moderate. Following Agapiou et al. (2016) and assuming a Gaussian prior distribution, $p_0(\theta) = \mathcal{N}(\mu, P)$, an effective dimension is defined by

$$\text{efd} = \text{Tr}\left( (P - \hat{P}) P^{-1} \right),$$

where $\hat{P}$ is the posterior covariance and where $\text{Tr}(A) = \sum_{j=1}^{n} a_{jj}$ is the trace of an $n \times n$ matrix $A$ with diagonal elements $a_{jj}$, $j = 1, \ldots, n$. Thus, the effective dimension measures the difficulty of a data assimilation problem by the differences between prior and posterior covariance. This means that the more information the data contains about the parameters, the higher is the problem's effective dimension and, thus, the harder is it to find the solution of the data assimilation problem. We emphasize that this is a statement about expected computational requirements and that it is counter-intuitive – parameters that are well-constrained by data should be easier to find than parameters that are mildly constrained by the data. However, in terms of computing or sampling posterior distributions, a high impact of data on parameter estimates makes the problem harder. Consider an extreme case where that data have no influence on parameter estimates. Then the posterior distribution is equal to the prior distribution and, thus, already known (no computations needed). If the data are very informative, the posterior distribution will be different from the prior distribution. For example, the prior may be "wide", i.e., not much is known about the parameters, while the posterior distribution is "tight", i.e., uncertainty in the parameters is small after the data are collected. Finding and sampling this posterior distribution requires significantly more (computational) effort than sampling the prior distribution.

Case (i) above is characterized by features that do not change (significantly) the posterior distribution and, hence, the features do not alter the effective dimension of the problem. It follows that the computed solutions and the required computational cost of the feature-based or "direct" approach are comparable. In case (ii) however, the feature changes the posterior distribution. Specifically, the dimension of the feature is lower than the dimension of the full data set because several important aspects of the data are neglected by the feature. A low-dimensional feature implies a low-dimensional feature-based likelihood, which in turn implies a low-dimensional feature-based posterior distribution. Since the feature neglects several aspects of the data, assimilating the feature will introduce a more gradual change from prior to posterior distribution than if all data are used. Thus, the feature-based approach reduces the effective dimension of the problem. For chaotic systems, this reduction in effective dimension can be so dramatic that the original problem is infeasible, while a feature-based approach becomes feasible, see Hakkarainen et al. (2012); Haario et al. (2015); Maclean et al. (2017) and section 4.4.

## 4    Numerical illustrations

We illustrate the above ideas with four numerical examples. In the examples, we also discuss appropriate numerical techniques for solving feature-based data assimilation problems. The first example illustrates that contributions from redundant data are negligible. The second example uses "real data" and a predator-prey model to illustrate the use of a PCA feature. Examples 1 and 2 are simple enough to solve by "classical" data assimilation, matching model outputs and data directly and serve as an illustration of problems of type (i) in section 3.3. Example 3 uses a low-dimensional model for a complex system, namely Earth's geomagnetic dipole field over the past 150 Myr. Here, a direct approach is infeasible, because the model and data are describing different scales and, thus, this example illustrates a problem of type (iii) (see section 3.3). Example 4 involves a chaotic partial differential equation (PDE) and parameter estimation is difficult using the direct approach because it requires estimating initial conditions from data. We design a robust feature that enables estimation of a parameter of the PDE without estimating initial conditions. The perturbed observation noise models for the features are successful in examples 1-3 and we use Monte Carlo for numerical solution of the feature-based problems. The perturbed observation method fails in example 4, which is also characterized by a noisy feature-based likelihood and we describe a different numerical approach based on maximum a posteriori estimates.

We wish to remind the reader that the choices of prior distributions are critical for the Bayesian approach to parameter estimation. However, the focus of this paper is on formulations of the likelihood and using features to define likelihoods. In the examples below we make reasonable choices for the priors, but other choices of priors will lead to different posterior distributions and, hence, different parameter estimates. In examples 1, 2 and 4, we do not have any information about the values of the parameters and we chose uniform priors over large intervals. In example 3, we use a sequential data assimilation approach and build priors informed by previous assimilations, as is typical in sequential data assimilation.

## 4.1 Example 1: more data is not always better

We illustrate that a data assimilation problem with fewer data points can be as useful as one with significantly more, but redundant data points. We consider a mass-spring-damper system

$$\frac{\mathrm{d}^2 x}{\mathrm{d}t^2} + 2\zeta\omega\frac{\mathrm{d}x}{\mathrm{d}t} + \omega^2 x = h(t-5),$$

where $t \geq 0$ is time, $\zeta > 0$ is a viscous damping coefficient, $\omega > 0$ is a natural frequency and $h(\tau)$ is the "step-function", i.e., $h(\tau) = 0$ for $\tau < 0$ and $h(\tau) = 1$ for $\tau \geq 0$. The initial conditions of the mass-spring-damper system are $x(0) = 0, \mathrm{d}x/\mathrm{d}t(0) = 0$. The parameters we want to estimate are the damping coefficient $\zeta$ and the natural frequency $\omega$, i.e., $\theta = (\zeta,\omega)^T$. To estimate these parameters we use a uniform prior distribution over the box $[0.5,4] \times [0.5,4]$ and measure the displacement $x(t)$ every $\Delta t = 0.5$ time units (starting with a measurement at $t = 0$). The duration of a (synthetic) experiment is $\tau = M\Delta t$ and we consider experiments of durations between $\tau = 25$ to $\tau = 250$ time units, with $M = 51$ to $M = 501$ data points. The data of an experiment of duration $\tau = M\Delta t$ are thus

$$z_i = x(i\Delta t) + v_i, \quad v_i \sim \mathcal{N}(0,10^{-3}), \quad i = 0,\ldots,M.$$

Writing $z = \{z_0,\ldots,z_M\}$, we obtain the likelihood

$$l_\tau(z|\theta) \propto \exp\left(-\frac{1}{2}\sum_{i=0}^{M} 10^3 \left(z_i - x(i\Delta T)\right)^2\right).$$

The likelihood and the uniform prior distribution define the posterior distribution

$$p_\tau(\theta|z) = \begin{cases} \frac{1}{C_\tau}l_\tau(z|\theta) & \text{if } \theta \in [0.5,4] \times [0.5,4], \\ 0 & \text{otherwise}, \end{cases}$$

where $C_M$ is a normalization constant. Data of an experiment of duration $\tau = 40$ is shown in figure 1(a). These synthetic data are generated with "true" parameters $\zeta = 1.5$ and $\omega = 1$. With these parameters the oscillator is "overdamped" and reaches its steady state ($\lim_{t\to\infty} x(t) = 1$) quickly. We anticipate that data collected after $t \approx 25$ is redundant in the sense that the same displacement is measured again and again. This suggests that the posterior distributions of experiments of duration $\tau = i\Delta t$ and $\tau = j\Delta t$ are approximately equal to each other, provided that $i,j > 50/\Delta t$. In other words, a data assimilation problem with $M = 101$ or $M = 251$ data points may have "roughly the same" posterior distribution and, consequently, lead to similar estimates.

We investigate this idea by solving data assimilation problems with experiment durations between $\tau = 25$ and $\tau = 225$. We compare the resulting posterior distributions $p_{25},\ldots,p_{225}$ to the posterior distribution $p_{250}$, corresponding to an experiment of duration $\tau = 250$. We use the Kullback Leibler (KL) divergence, $D_{\mathrm{KL}}(\hat{p}_0||\hat{p}_1)$ of two distributions to measure "how far" two distributions are from one another. For two $k$-dimensional Gaussians $p_0 = \mathcal{N}(m_0, P_0)$ and $p_1 = \mathcal{N}(m_1, P_1)$, the KL divergence is given by

$$D_{\mathrm{KL}}(p_0||p_1) = \frac{1}{2}\left(\mathrm{Tr}(P_1^{-1}P_0) + (m_1 - m_0)^T P_1^{-1}(m_1 - m_0) - k + \log\left(\frac{\det P_1}{\det P_0}\right).\right)$$

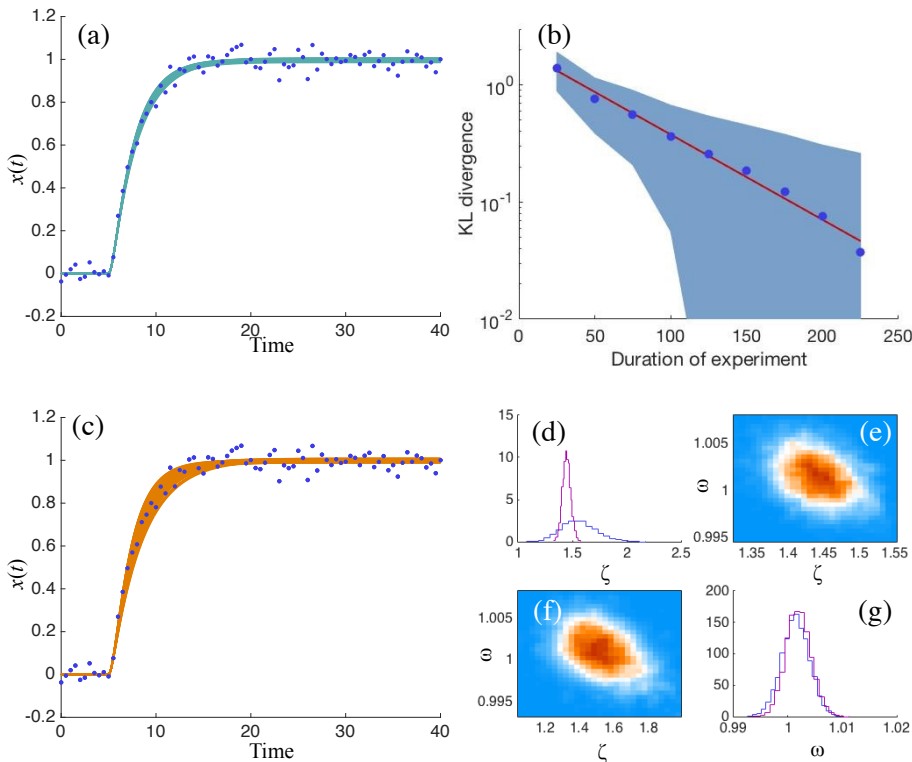

**Figure 1.** (a) Data $z_i$, $i = 1, \ldots, 80$ (blue dots) of an experiment of duration $\tau = 40$ and 50 trajectories of oscillators with damping coefficient and natural frequency drawn from the posterior distribution $p(\theta|z)$ (turquoise). (b) KL divergence of approximate posterior distributions $D_{\text{KL}}(\hat{p}_{250}||\hat{p}_\tau)$, $M = 25, \ldots, 225$, as a function of the duration $\tau$ of an experiment. Blue dots – average KL divergence of 1000 experiments. Red line – exponential fit. Light blue cloud: confidence interval based on standard deviations observed during the 1000 experiments. (c) Same data as in (a) (blue dots) and 50 trajectories of oscillators with damping coefficient and natural frequency drawn from the feature-based posterior distribution. (d) Histogram of the marginal $p_{40}(\zeta|z_{80})$ of the posterior distribution $p_{40}(\theta|z_{80})$ (purple) and histogram the marginal $p_{\mathcal{F}}(\xi|f)$ of the feature-based posterior distribution $p_{\mathcal{F}}(\theta|f)$ (blue). (e) Two-dimensional histogram of the posterior distribution $p_{40}(\theta|z_{80})$. (f) Two-dimensional histogram of the feature-based posterior distribution $p_{\mathcal{F}}(\theta|f)$. (g) Histogram of the marginal $p_{40}(\omega|z_{80})$ of the posterior distribution $p_{40}(\theta|z_{80})$ (purple) and histogram the marginal $p_{\mathcal{F}}(\omega|f)$ of the feature-based posterior distribution $p_{\mathcal{F}}(\theta|f)$ (blue).

Note that $D_{\text{KL}}(p_0||p_1) = 0$ if the two distributions are identical and a large $D_{\text{KL}}(p_0||p_1)$ suggest that $p_0$ and $p_1$ are quite different. Computing the KL divergence for non-Gaussian distributions is numerically more challenging and here were are content to measure the distance of two distributions by the KL divergence of their Gaussian approximations. We thus compute Gaussian approximations to the posterior distributions $p_{25}, \ldots, p_{250}$, by computing the posterior mode $\theta^*$ (by Gauss-Newton optimization) and the Hessian $H$ of the negative logarithm of the posterior distribution at the mode. We then define the Gaussian approximation by

$$p_\tau(\theta|z_M) \approx \hat{p}_\tau(\theta|z_M) = \mathcal{N}(\theta^*, H^{-1}), \tag{8}$$

and use $D_{\mathrm{KL}}(\hat{p}_{250}||\hat{p}_\tau)$ to measure the distance of $p_{250}$ and $p_\tau$.

Each experiment is in itself a random event because the measurement noise is random. The KL divergence between the various posterior distributions is, thus, also random and we address this issue by performing 1000 independent experiments and then average the KL divergences. Our results are shown in figure 1(b). We plot the average KL divergence, as well as "error bars" based on the standard deviation, as a function of the experiment duration and note an exponential decrease of KL divergence with experiment duration or, equivalently, number of data points used for parameter estimation. Thus, as we increase the number of data points, the posterior distributions get closer, as measured by this KL divergence, to the posterior distribution with $M = 501$ data points. In other words, we obtain very similar posterior distributions with $M = 101$ or $M = 501$ data points. This indicates that the steady state data can be neglected because there is little additional information in these data. These results suggest that the data can be compressed without significant loss of information about the parameters. One could, for example, define a feature by simply neglecting data collected after $t > 30$. This feature would lead to almost identical parameter estimates as using the full data set.

We now consider a feature that compresses the data into two numbers. The first component of our feature is the average of the last 50 data points. This average is directly related to the natural frequency since $\lim_{t \to \infty} x(t) = 1/\omega^2$. The second component of the features is the slope of a linear fit to the seven data points collected after $t = 5$, i.e., after the step is applied.

The covariance matrix $R$ of the assumed Gaussian noise $\eta$ (see equation 6), using the perturbed observation approach as described in section 3.1. We generate $10^3$ perturbed data sets to compute $R$ and find that the off-diagonal elements are small compared to the diagonal elements. We thus neglect the correlation between the two components of the feature, but this is not essential. Altogether the feature-based likelihood is given by

$$l_{\mathcal{F}}(f|\theta) \propto \exp\left(-\frac{1}{2}\left(f - \mathcal{F}_{\mathcal{M}}(\theta)\right)^T R^{-1}\left(f - \mathcal{F}_{\mathcal{M}}(\theta)\right)\right),$$

where $\mathcal{F}_{\mathcal{M}}$ represents the computations (*i*) simulate the oscillator with parameters $\theta$ for $\tau$ time units; and (*ii*) compute the feature, i.e., the average steady state value and slope, as described above. Together with the uniform prior distribution, we obtain the feature-based posterior distribution

$$p_{\mathcal{F}}(\theta|f) = \begin{cases} \frac{1}{C_{\mathcal{F}}} l_{\mathcal{F}}(f|\theta) & \text{if } \theta \in [0.5, 4] \times [0.5, 4], \\ 0 & \text{otherwise,} \end{cases}$$

where $C_{\mathcal{F}}$ is a normalization constant.

We solve this feature-based problem for an experiment of duration $\tau = 40$ by implicit sampling (see section 2.2) using $N_e = 10^3$ samples. From these samples we compute $\rho \approx 1.07$, i.e., almost all samples are effective samples. Results are illustrated in figure 1(c), where we plot trajectories corresponding to 50 samples of $\theta = (\zeta, \omega)$ of the feature-based posterior distribution. We note that the trajectories are all "near" the data points. For comparison, we also solve the data assimilation problem without using features and compute $p_{40}$ (see equation (8)), also by implicit sampling with $N_e = 10^3$ samples. We find that $\rho \approx 1.38$ in this case. We note that the feature-based posterior distribution is different from the "classical" one. This can be seen by comparing the clouds of trajectories in figures 1(a) and 1(c). The wider cloud of trajectories indicates that the feature does

not constrain the parameters as much as the full data set. The relaxation induced by the feature-based approach however also results in the feature-based approach being slightly more effective in terms of the number of effective samples.

Finally, we show triangle-plots of the posterior distribution $p_{40}$ and the feature-based posterior distribution in figures 1(d)-(g). A triangle plot of the feature-based posterior distribution $p_{\mathcal{F}}$ consists of histograms of the marginals $p_{\mathcal{F}}(\zeta|f)$ and $p_{\mathcal{F}}(\omega|f)$,
plotted in blue in figures 1(d) and 1(e), and a histogram of $p_{\mathcal{F}}(\theta|f)$ in figure 1(f). A triangle plot of the posterior distribution $p_{40}(\theta|z_{80})$ is shown in figures 1(d), (e) and (f). Specifically, we plot histograms of the marginals $p_{40}(\zeta|z_{80})$ and $p_{40}(\omega|z_{80})$ in purple in figures 1(d) and 1(g) and we plot a histogram of the posterior distribution $p_{40}(\theta|z_{80})$ in figure 1(e). We find that the marginals $p_{\mathcal{F}}(\omega|f)$ and $p_{40}(\omega|z_{80})$ are nearly identical, which indicates that the feature constrains the frequency $\omega$ nearly as well as the full data set. The damping coefficient $\zeta$ is less tightly constrained by our feature, which results in a wider posterior
distribution $p_{\mathcal{F}}(\zeta|f)$ than $p_{40}(\zeta|z_{80})$. A more sophisticated feature that describes the transient behavior in more detail would lead to different results, but our main point is to show that even our simple feature, which neglects most of the data, leads to useful parameter estimates.

## 4.2  Example 2: predator-prey dynamics of lynx and hares

We consider the Lotka-Volterra (LV) equations (Lotka, 1926; Volterra, 1926)

$$\frac{\mathrm{d}x}{\mathrm{d}t} = \alpha x - \beta x y, \quad \frac{\mathrm{d}y}{\mathrm{d}t} = -\gamma y + \delta x y,$$

where $t$ is time, $\alpha, \beta, \gamma, \delta > 0$ are parameters and $x$ and $y$ describe "prey" and "predator" populations. Our goal is to estimate the four parameters in the above equations as well as the initial conditions $x_0 = x(0)$, $y_0 = y(0)$, i.e., the parameter vector we consider is $\theta = (\alpha, \beta, \gamma, \delta, x_0, y_0)^T$. Since we do not have prior information about the parameters, we chose a uniform prior distribution over the six-dimensional cube $[0, 10]^6$.

We use the lynx and hare data of the Hudson's Bay Company (Gilpin, 1973; Leigh, 1968) to define a likelihood. The data set covers a period from 1897 to 1935, with one data point per year. Each data point is a number of lynx furs and hare furs, with the understanding that the number of collected furs is an indicator for the overall lynx or hare population. We use data from 1917 to 1927, because the solution of the LV equations is restricted to cycles of fixed amplitude and the data during this time period roughly has that quality. We scale the data to units of "$10^4$ hare furs" and "$10^3$ lynx furs" (so that all numbers are order one).
We use this classical data set here, but predator-prey models have recently also been used in low-dimensional cloud models that can represent certain aspects of large eddy simulations (Koren and Feingold, 2011; Feingold and Koren, 2013). However, the sole purpose of this example is to demonstrate that the feature-based approach is robust enough for use with "real" data (rather than the synthetic data used in example 1).

We define a feature $f$ by the first (largest) singular value and the first left and right singular vectors of the data. The feature
vector $f$ thus has dimension 14 (we have $2 \times 11$ raw data points). We compute the noise $\eta$ for the feature-based likelihood using the "perturbed observation" method as above. We generate 10,000 perturbed data sets by adding realizations of a Gaussian random variable with mean of zero and unit covariance to the data. The resulting sample covariance matrix serves as the matrix

$R_f$ in the feature-based likelihood. Note that our choice of noise on the "raw" data is somewhat arbitrary. However, as stated above, the main purpose of this example is to demonstrate our ideas, not to research interactions of lynx and hare populations.

We use the MATLAB implementation of the affine invariant ensemble sampler to solve the feature-based data assimilation problem, see Grinsted (2017); Goodman and Weare (2010). We use an ensemble size $N_e = 12$ and each ensemble member produces a chain of length $n_s = 8334$. We thus have $N = 100,008$ samples. Each chain is initialized as follows: we first find the posterior mode using Gauss-Newton optimization. To do so, we perform an optimization with different starting points and then chose the optimization result that leads to the largest feature-based posterior probability. The initial values for our ensemble of walkers are twelve draws from a Gaussian distribution whose mean is the posterior mode and whose covariance is a diagonal matrix with elements $(0.02, 0.02, 0.02, 0.02, 0.2, 0.2)$. We disregard the first 2,500 steps of each chain as "burn-in" and compute an average IACT of 735, using the methods described in Wolff (2004). We have also performed experiments with larger ensembles ($N_e = 12$ is the minimum ensemble size for this method), and with different initializations of the chains and obtained similar results. We have also experimented with the overall number of samples (we used up to $10^6$ samples) and obtained similar results.

We show a triangle-plot of the feature-based posterior distribution, consisting of histograms of all one- and two-dimensional marginals, in figure 2. We observe that there is strong correlation between the parameters $\alpha, \beta, \gamma, \delta$, but less so between these parameters and the initial conditions. A reason for the strong correlations between the parameters is that only combinations of the parameters define the solution of the differential equation (after non-dimensionalization). Perhaps most importantly, we find that the feature-based posterior distribution constraints the parameters well, especially compared to the prior distribution which is a hyper-cube with sides of length ten.

We plot the trajectories of the LV equations corresponding to 100 samples of the feature-based posterior distribution in figure 3. We note that the trajectories pass near the 22 original data points (shown as orange dots in figure 3). The fit of the lynx population is particularly good, but the trajectories of the hare populations do not fit the data well. For example, all model trajectories bend downwards towards the end of the cycle, but the data seem to exhibit an upward tendency. However, this inconsistency is not due to the feature-based approach. In fact, we obtain similar solutions with a "classical" problem formulation. The inconsistency is due to the limitations of the LV model, which is limited to cycles, whereas the data are not cyclic. Nonetheless, our main point here is that the feature-based approach is sufficiently robust that it can handle "real" data and "simple" models. We also emphasize that this data assimilation problem is not difficult to do by the "classical" approach, i.e., without using features. This suggests that this problem is of category (i) in section 3.3.

### 4.3 Example 3: variations in Earth's dipole's reversal rates

We consider Earth's magnetic dipole field over time-scales of tens of millions of years. On such time-scales, the geomagnetic dipole exhibits reversals, i.e., the north pole becomes the south pole and vice versa. The occurrence of dipole reversals is well documented over the past 150 Myr by the "geomagnetic polarity time scale" (Cande and Kent, 1995; Lowrie and Kent, 2004) and the dipole intensity over the past 2 Myr is documented by the Sint-2000 and PADM2M data sets (Valet et al., 2005; Ziegler et al., 2005). Several low-dimensional models for the dipole dynamics over the past 2 Myr have been created see, e.g., Hoyng

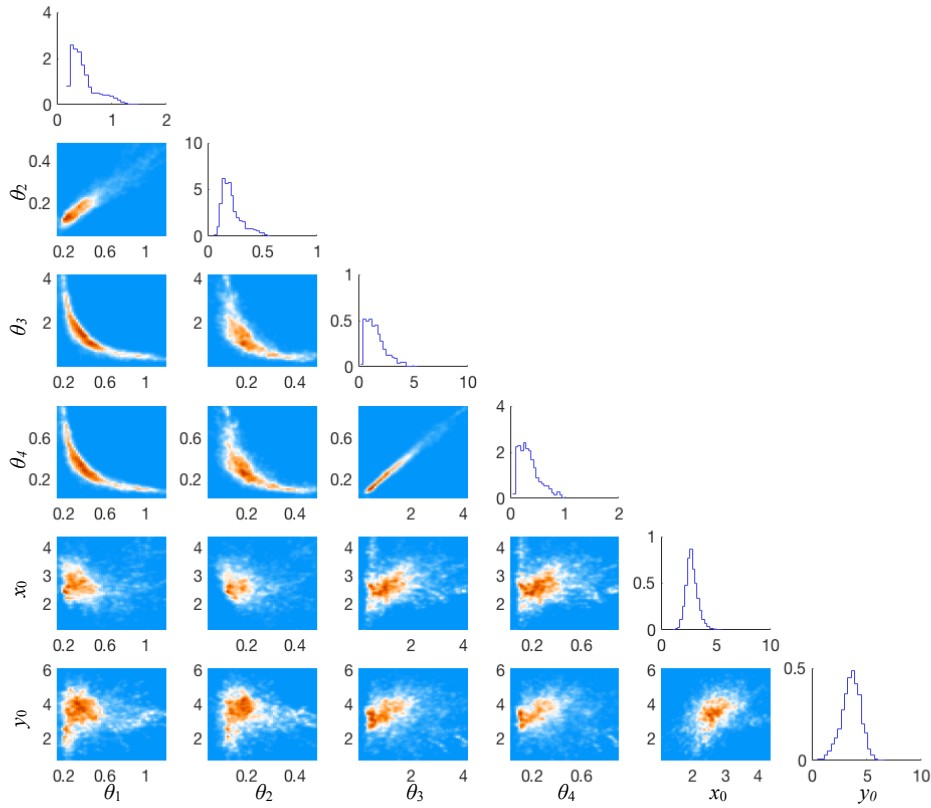

**Figure 2.** Triangle plot of histograms of all one and two-dimensional marginals of the feature-based posterior distribution.

et al. (2005); Brendel et al. (2007); Kuipers et al. (2009); Buffett et al. (2014); Buffett and Matsui (2015). We consider two of these models and call the model of Petrelis et al. (2009) the P09 model and the one of Buffett et al. (2013) the B13 model. The B13 model is the stochastic differential equation (SDE)

$$\mathrm{d}x = f(x)\mathrm{d}t + g(x)\mathrm{d}W, \tag{9}$$

5    where $t$ is time in Myr, $x$ describes the dipole intensity and where $W$ is Brownian motion (see Buffett et al. (2013) for details). The functions $f$ and $g$ are called the drift- and diffusion coefficients and in Buffett et al. (2013), $f$ is a spline and $g$ a polynomial whose coefficients are computed using PADM2M. We use the same functions $f$ and $g$ as described in Buffett et al. (2013). The P09 model consists of an SDE of the form (9) for a "phase", $x$, with $f(x) = \alpha_0 + \alpha_1 \sin(2x)$, $g(x) = 0.2\sqrt{|\alpha_1|}$, $\alpha_1 = -185\,\mathrm{Myr}^{-1}$, $\alpha_0/\alpha_1 = -0.9$ and $\theta(t) = 1$. The dipole is computed from the phase $x$ as $D = R\cos(x + x_0)$, where $x_0 = 0.3$
10   and $R = 1.3$ defines the amplitude of the dipole.

    In both models, the drift, $f$, represents known, or "resolved" dynamics and the diffusion coefficient $g$, along with Brownian motion $W$, represents the effects of turbulent fluid motion of Earth's liquid core. The sign of the dipole variable defines the dipole polarity. We take the negative sign to mean "current configuration" and a positive sign means "reversed configuration".

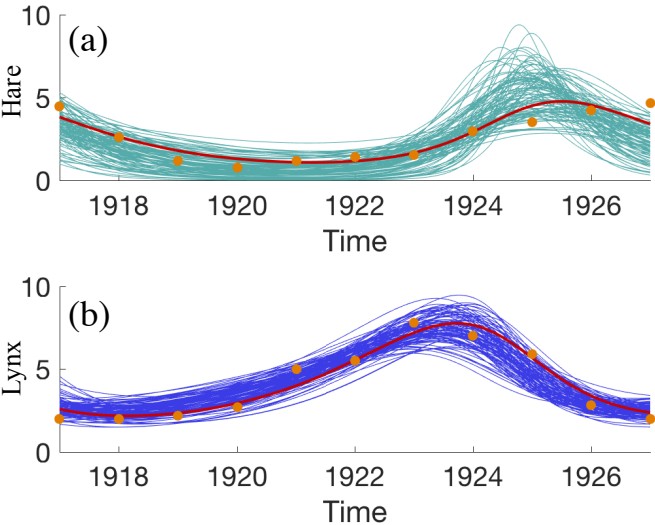

**Figure 3.** Raw data (orange dots), trajectories corresponding to the feature-based posterior mode (red) and 100 trajectories of Hares (turquoise) in (a) and Lynx (blue) in (b), corresponding to 100 samples of the feature-based posterior distribution.

A period during which the dipole polarity is constant is called a "chron". The P09 and B13 models exhibit chrons of varying lengths, however the mean chron duration (MCD) is fixed. With the parameters cited above the models yield an MCD on the same order of magnitude as the one observed over the past 30 Myr. Simulations of the B13 and P09 model are illustrated in figure 4, where we also show the last 100 Myr of the geomagnetic polarity time scale.

The geomagnetic polarity time scale shows that Earth's MCD varies over the past 150 Myr. For example, there were 125 reversals between today and 30.9 Myr ago (MCD $\approx 0.25$ Myr) , 57 reversals between 30.9 Myr ago and 73.6 Myr ago (MCD $\approx$ 0.75 Myr), and 89 between 120.6 Myr ago and 157.5 Myr (MCD $\approx 0.41$ Myr) (Lowrie and Kent, 2004). The B13 and P09 models exhibit a constant MCD and, therefore, are valid over periods during which Earth's MCD is also constant, i.e., a few million years. We modify the B13 and P09 models so that their MCD can vary over time, which makes the models valid

for periods of more than 100 Myr. The modification is a time-varying, piecewise constant parameter $\theta(t)$ that multiplies the diffusion coefficients of the models. The modified B13 and P09 models are thus SDEs of the form

$$\mathrm{d}x = f(x)\mathrm{d}t + \theta(t)g(x)\mathrm{d}W. \tag{10}$$

We use feature-based data assimilation to estimate the value of $\theta(t)$ such that the modified B13 and P09 models exhibit similar MCDs as observed in the geomagnetic polarity time-scale over the past 150 Myr. Note that straightforward application of data

assimilation is not successful in this problem. We tried several particle filters to assimilate the geomagnetic polarity time scale more directly into the modified B13 and P09 models. However, we had no success with this approach because the data contain only information about the sign of the solution of the SDE.

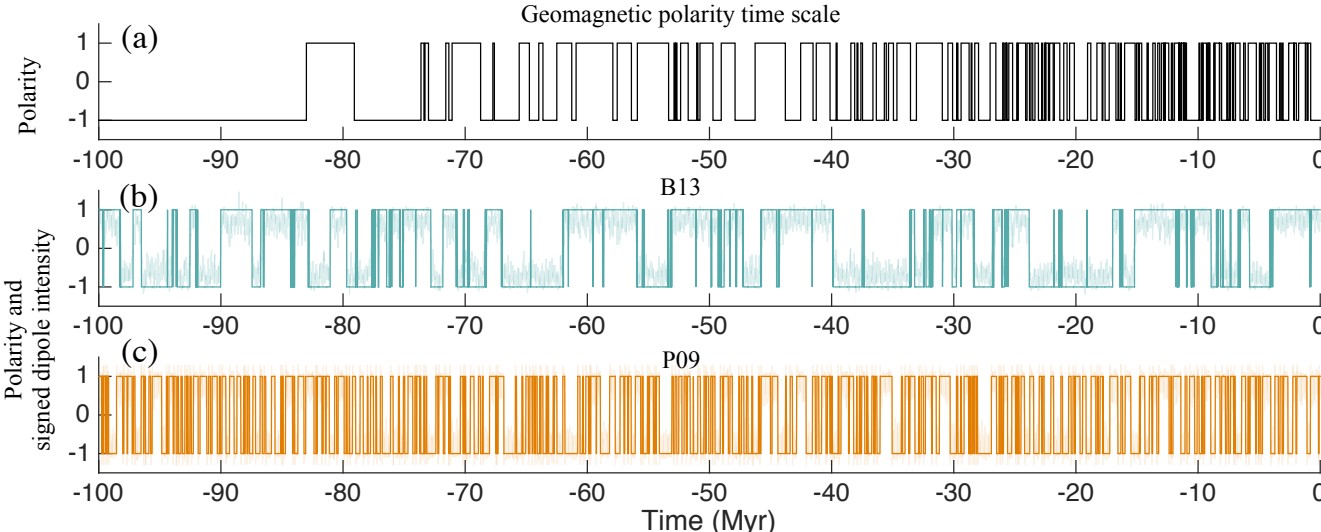

**Figure 4.** (a) Earth's dipole polarity over the past 100 Myr (part of the geomagnetic polarity time scale). (b) A 100 Myr simulation with B13 and the associated sign function. (c) A100 Myr simulation with P09 and the associated sign function.

The feature we extract from the geomagnetic polarity time scale is the MCD, which we compute by using a sliding window average over 10 Myr. We compute the MCD every 1 Myr, so that the "feature data", $f_1, \ldots, f_{149}$, are 149 values of MCD. We obtain these 149 values by using the geomagnetic polarity time scale (starting at 157.53 Myr ago) and a 10 Myr averaging window. For the first data point, $f_1$, we use slightly less than 10 Myr of data (from 157.53 Myr ago to 148 Myr ago). The aver-
aging window is always "left to right", i.e., we average from the past to the present. For the last few data points ($f_{144} \ldots f_{148}$), the averaging is not centered and uses 10 Myr of data "to the left".

The geomagnetic polarity time scale and the MCD feature are shown in figure 5. We note that the averaging window of 10 Myr is too short during long chrons, especially during the "cretaceous superchron" that lasted almost 40 Myr (from about 120 to 80 Myr ago). We set the MCD to be 250 Myr whenever no reversal occurs within our 10 Myr window. This means that
the MCD feature has no accuracy during this time period, but indicates that the chrons are long.

To sequentially assimilate the feature data, we assume that the parameter $\theta(t)$ is piecewise constant over 1 Myr intervals and estimate its value $\theta_k = \theta(k \cdot 1\text{Myr})$, $k = -147, \ldots, 0$ based on the feature $f_k$ and our estimate of $\theta_{k-1}$. The feature $f_k$ and the modified B13 and P09 models are connected by the equation

$$f_k = \mathcal{M}_{\mathcal{F}}(\theta_k) + \eta_k, \tag{11}$$

which defines the feature-based likelihood and where $\mathcal{M}_{\mathcal{F}}$ are the computations required to compute the MCD for a given $\theta_k$. These computations work with a discretization of the modified P09 and B13 SDEs using a 4th-order Runge-Kutta scheme for the deterministic part ($f\mathrm{d}t$), and an Euler-Maruyama scheme of the stochastic part ($\theta_k g(x)\mathrm{d}W$). The time step is 1 kyr. For a given $\theta_k$, we perform a simulation for a specified number of years and compute MCD based on this run. All simulations

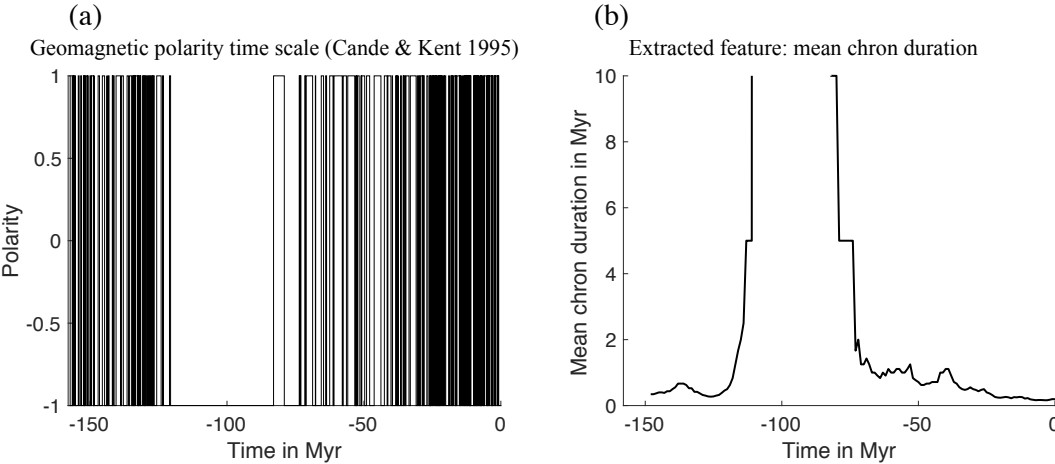

**Figure 5.** (a) Geomagnetic polarity time scale. (b) MCD, averaged over a 10 Myr window, every 1 Myr.

are initialized with zero initial conditions (but the precise value of the initial conditions is not essential because it is averaged out over the relatively long simulations) and are performed with a fixed value for $\theta_k$. The value of $\theta_k$ determines the duration of a simulation, since small values of $\theta_k$ require longer simulations because the chrons tend to become longer. Specifically, we perform a simulation of 300 Myr if $\theta_k < 0.7$, of 100 Myr if $0.7 \leq \theta_k < 1$, of 50 Myr if $1 \leq \theta_k < 1.6$ and of 20 Myr if

$\theta_k \geq 1.6$. Note that computation of MCD, in theory, requires an infinite simulation time. We chose the above simulation times to balance a computational budget, while at the same time our estimates of MCD are reliable enough to avoid large noise during feature-based likelihood evaluations.

For the modified B13 model we add one more step. The numerical solutions of this model tend to exhibit short chrons (a few thousand years) during a "proper reversal", i.e., when the state transitions from one polarity (+1) to the other (-1), it crosses

zero several times. On the time scales we consider, such reversals are not meaningful and we filter them out by smoothing the numerical solutions of the modified B13 model by a moving average over 25 kyrs. In this way, the chrons we consider and average over have a duration of at least tens of thousands of years.

We investigate how to chose the random variable $\eta$ in (11), which represents the noise in the feature, by performing extensive computations. For each model (B13 and P09), we chose a grid of $\theta$ values that lead to MCD that we observe in the geomagnetic

polarity time scale. The $\theta$-grid is different for the B13 and P09 model because the dependency of MCD on $\theta$ is different for both models and because computations with P09 are slightly faster. For both models, a small $\theta$ leads to reversal being rare, even during 300 Myr simulations. We chose to not consider $\theta$ smaller than $0.3$, again for computational reasons and because, as explained above, our simulations and computations lose accuracy during very long chrons such as the cretaceous superchron. Thus, the "actual" $\theta$ during a period with large MCD may be smaller then the lower bound we compute, however we cannot

extract that information from the feature data and the computational framework we construct. This means that if the upper or lower bounds of $\theta$ are achieved, all we can conclude is that $\theta$ is small (large), perhaps smaller (larger) than our assumed lower

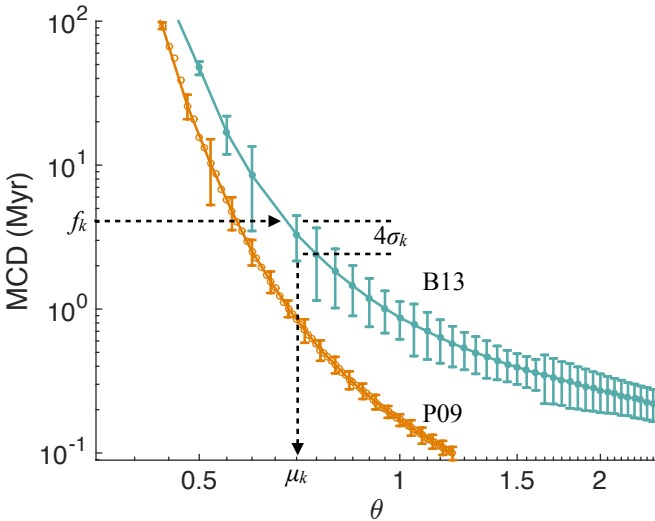

**Figure 6.** MCD as a function of $\theta$ for the B13 model (turquoise) and the P09 model (orange). Shown are the average MCD (solid lines) and two-standard-deviation error bards computed from 100 simulations. This graph is used to define the standard deviation of the feature-noise $\eta_k$ as well as the mean of the proposal distribution $q_k$. For the P09 model, we plot the standard deviations only for every other $\theta$ value for readability.

(upper) bound, which leads to MCDs that are longer (smaller) than what we can actually compute with our model and bounded model parameters.

For each value of $\theta$ on our grid, we perform 100 simulations and, for each run, compute average MCD. The mean and standard deviation of average MCD, computed from these simulations, are shown in figure 6. We occasionally observe large standard deviations for small $\theta_k$, because only a few reversals may occur during these runs, which makes estimates of the standard deviations unreliable (see above). In this case, we assign a maximum standard deviation of 2.5 Myr. We base our feature-error model $\eta_k$ on this graph and pick $\eta_k$ to be a zero mean Gaussian with a standard deviation $\sigma_k$ that we read from the graph as illustrated by figure 6, i.e., for a given $f_k$, we use the standard deviation we computed for the nearest point on our MCD-$\theta$-grid.

A feature $f_k$ defines $\eta_k$ and then equation (11) defines a feature-based likelihood. We define a prior distribution by the Gaussian $p_{0,k}(\theta_k) = \mathcal{N}(\bar{\theta}_{k-1}, \sigma_0^2)$, where $\sigma_0 = 0.1$ and where $\bar{\theta}_{k-1}$ is the mean value we computed at the previous time, $k-1$ (we describe what we did for the first time step $k = 1$ below). This results in the feature-based posterior

$$p_k(\theta_k | f_k) \propto \exp\left( -\frac{1}{2\sigma_k^2} \left(f_k - \mathcal{M}_{\mathcal{F}}(\theta_k)\right)^2 - \frac{1}{2\sigma_0^2} \left(\bar{\theta}_{k-1} - \theta_k\right)^2 \right).$$

We draw 100 samples from this posterior distribution by direct sampling with a proposal distribution $q_k(\theta_k) = \mathcal{N}(\mu_k, \sigma_q)$, where $\sigma_q = 0.05$ and where $\mu_k$ is based on the MCD-$\theta$ graph shown in figure 6, i.e., we chose $\mu_k$ to be the $\theta$-value corresponding to the MCD value $f_k$ we observe. We have experimented with other values of $\sigma_q = 0.05$ and found that how $\sigma_q$ is chosen

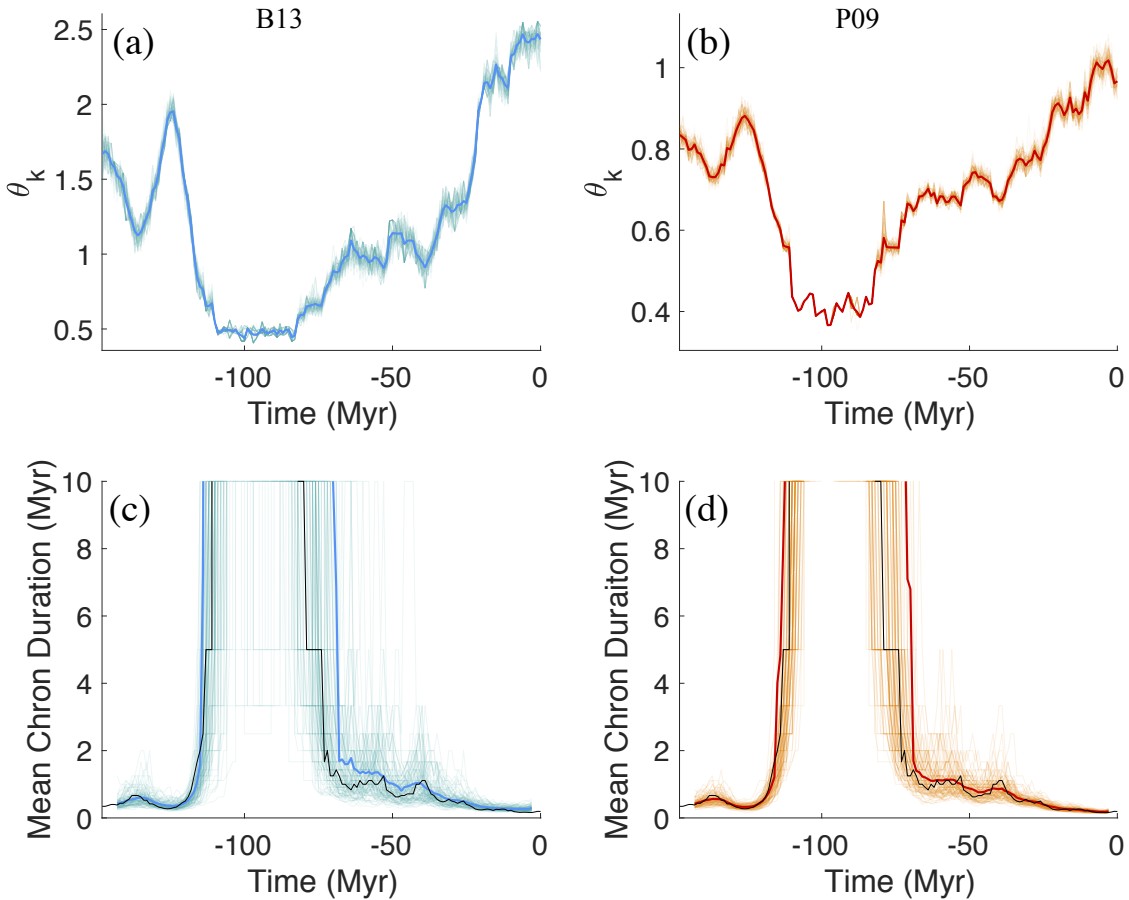

**Figure 7.** (a) $\theta_k$ as a function of time for modified B13; 100 samples of feature-based posterior distributions $p_k(\theta_k|f_k)$ (light turquoise) and their mean (blue). (b) $\theta_k$ as a function of time for modified P09; 100 samples of feature-based posterior distributions $p_k(\theta_k|f_k)$ (light orange) and their mean (red). (c) features $f_k$ computed by drawing 100 samples (light turquoise) from the feature-based posterior distribution of the modified B13 model and their mean (blue). (d) features $f_k$ computed by drawing 100 samples (light orange) from the feature-based posterior distribution of the modified P09 model and their mean (red). The MCD feature extracted from the geomagnetic polarity time scale is shown in black.

is not critical for obtaining the results we present. We repeat this process for all but the very first of the features $f_k$. For the first step, $k = 1$, we set the prior distribution equal to the proposal distribution.

Our results are illustrated in figure 7. Figures 7(a) and 7(b) show 100 samples of the posterior distributions $p_k(\theta_k|f_k)$ as a function of time, as well as their mean. The panel on the right shows results for the modified B13 model, the panel on the left shows results for the modified P09 model. We note that, for both models, $\theta_k$ varies significantly over time. The effect that a time-varying $\theta$ has on the MCD of the modified B13 and P09 models is illustrated in figures 7(c) and 7(d), where we plot 100 features generated by the modified P09 and B13 models using the 100 posterior values of $\theta_k$ shown in the top row. We

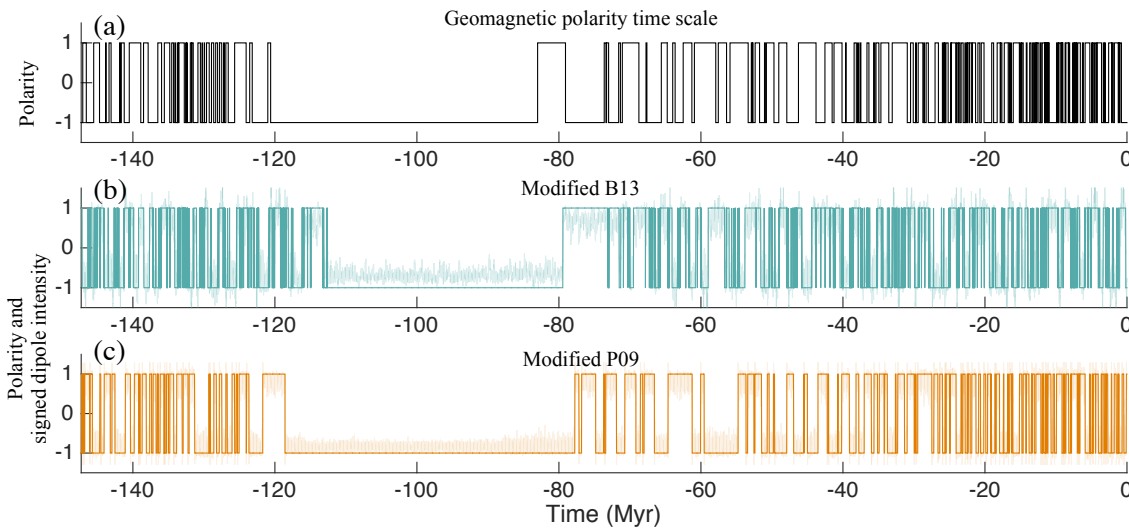

**Figure 8.** (a) Geomagnetic polarity time scale. (b) Modified B13 model output with $\theta_k$ drawn from the feature-based posterior distributions. (c) Modified P09 model output with $\theta_k$ drawn from the feature-based posterior distributions.

note a good agreement with the recorded feature (shown in black). This is perhaps not surprising, since we use the feature data to estimate parameters, which in turn reproduce the feature data. However, this is a basic check that our data assimilation framework produces meaningful results.

We further illustrate the results of the feature-based data assimilation in figure 8, where we plot the geomagnetic polarity time scale as well as the dipole of the modified B13 and P09 models, generated by using a sequence $\theta_k$, drawn from the feature-based posterior distributions. We note that the modified models exhibit a time-varying MCD and that superchrons (chrons longer than 10 Myr) appear in both models at (roughly) the same time as on Earth.

The advantage of the feature-based approach in this problem is that it allows us to calibrate the modified B13 and P09 models to yield a time-varying MCD in good agreement with the data (geomagnetic polarity time-scale), where "good agreement" is to be interpreted in the feature-based sense. Our approach may be particularly useful for studying how flow structure at the core affects the occurrence of superchrons. A thorough investigation of what our results imply about the physics of geomagnetic dipole reversals will be the subject of future work. In particular, we note that other choices for the standard deviation $\sigma_0$, that defines expected errors in the feature, are possible and that other choices will lead to different results. If one wishes to use the feature-based approach presented here to study Earth's deep interior, one must carefully chose $\sigma_0$. Here we are content with showing how to use feature-based data assimilation in the context of geomagnetic dipole modeling.

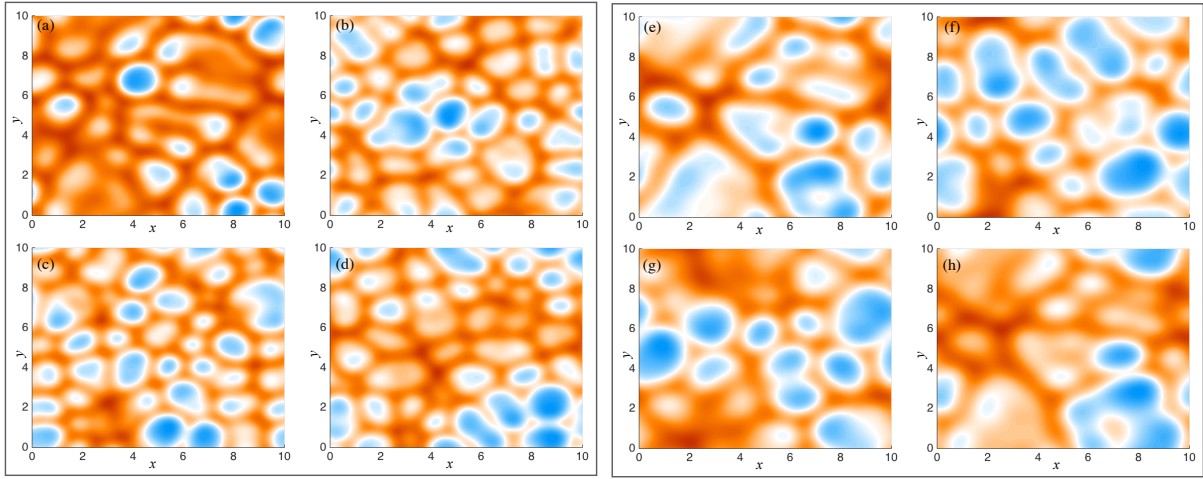

**Figure 9.** (a)-(d) Four snapshots of the solution of the KS equation with $\theta = 1.55$. (e)-(g) Four snapshots of the solution of the KS equation with $\theta = 3.07$.

### 4.4 Example 4: parameter estimation for a Kuramoto-Sivashinsky equation

We consider the Kuramoto-Sivashinksy (KS) equation

$$\frac{\partial \phi}{\partial t} = -\theta \nabla^2 \phi - \nabla^4 \phi + |\nabla \phi|^2,$$

where $t \in [0, T]$, the spatial domain is a two-dimensional square $[x, y] \in [0, 10\pi] \times [0, 10\pi]$ and the boundary conditions are periodic. Here $\nabla = (\partial/\partial x, \partial/\partial y)$ and $\theta$ is the parameter we want to estimate. We use a uniform prior distribution over $[0, 5]$. As in earlier examples, our focus is on formulating likelihoods and our choice of prior is not critical to the points we wish to make when illustrating the feature-based techniques. The initial condition of the KS equation is a Gaussian random variable, which we chose as follows. We simulate the KS equation for $T$ "time" units starting from uniformly distributed Fourier coefficients within the unit-hypercube (see a few sentences below for how these simulations are done). We pick $T$ large enough so that $\phi(x, y, T)$ varies smoothly in space. We repeat this process 100 times to obtain 100 samples of solutions of the KS equation. The resulting sample mean and sample covariance matrix of the solution at time $T$ define the mean and covariance of the Gaussian which we use as a random initial condition below.

For computations we discretize the KS equation by the spectral method and exponential time differencing with $\delta t = 0.005/\theta$. For a given $\theta$, we then compute $\phi$ in physical space by Fourier transform and interpolation onto a $256 \times 256$ grid. The solution of the KS equation depends on the parameter $\theta$ in a way that a typical spatial scale of the solution, i.e., the scale of the "valleys and hills" we observe, increases as $\theta$ decreases, as illustrated by figure 9, where we show snapshots of the solution of the KS equation after 2500 time steps for different two different choices of the parameter $\theta$.

The data are 100 snapshots of the solution of the KS equation obtained as follows. For a given $\theta$, we draw an initial condition from the Gaussian distribution (see above) and simulate for 2500 time steps. We save the solution on the $256 \times 256$ grid every

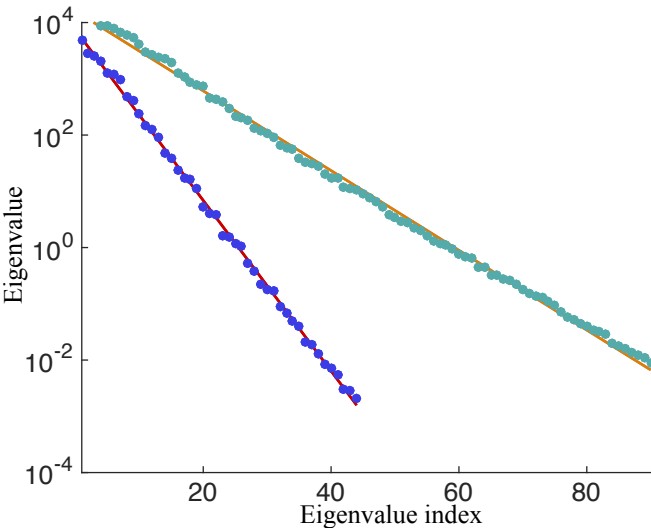

**Figure 10.** Illustration of the computed feature. Eigenvalues of covariance matrices of snapshots (dots) and log-linear fit (solid lines). Blue dots and red line correspond to a run with $\theta = 1.55$, turquoise dots and orange line correspond to a run with $\theta = 3.07$.

50 time steps. We repeat this process, with another random initial condition drawn from the same Gaussian distribution, to obtain another 50 snapshots of the solution. The 100 snapshots constitute a data set with a total number of more than 6 million points.

5 The feature we extract from the data is as follows. We interpolate the snapshots onto a coarser $64 \times 64$ grid and use the coarsened snapshots to compute a covariance matrix. Then we compute the largest eigenvalues of the covariance matrix and compute a linear approximation to the logarithm of the eigenvalues (using least squares). The slope and intercept of this line constitutes the feature. An example is shown in figure 10. We chose this features because the parameter $\theta$ defines the spatial scale of the solution (see above) and this scale is connected to the length-scale of a covariance function of a Gaussian process approximation of the solution. The length scale of the Gaussian process in turn defines the exponential decay of the eigenvalues

10 of its associated covariance matrix and this decay is what we capture by our feature. In simple terms, the larger the length-scale, the faster is the decay of the eigenvalues.

 It is important to note that the feature we construct does not depend on the initial conditions. This is the main advantage of the feature-based approach. Using the feature, rather than the trajectories, enables estimation of the parameter $\theta$ *without* estimation of initial conditions. With a likelihood based on the mismatch of model and data, one has to estimate the parameter

15 $\theta$ *and* the initial conditions, which makes the effective dimension of the problem large, so that the required computations are substantial. Most importantly, estimating the initial condition based on a mismatch of model output and data is difficult because the KS equation is chaotic. For these reasons, the feature-based approach makes estimation of the parameter $\theta$ feasible. Note that the feature has also reduced the effective dimension of the problem (see section 3.3.4) because the number of parameters

to be estimated has been reduced from the number of modes ($256^2$) to one. The price to be paid for this reduction in (effective) dimension is that the feature-based approach does not allow us to compute trajectories that match the data point-wise.

The feature-based likelihood is defined by the equation

$$f = \mathcal{M}_{\mathcal{F}}(\theta) + \eta, \quad \eta \sim \mathcal{N}(0, R), \tag{12}$$

where $f = \mathcal{F}(z)$ is the feature computed from the data, $R$ is a $2 \times 2$ covariance matrix (see below) and where $\mathcal{M}_{\mathcal{F}}$ is shorthand for the following computational steps for a given parameter $\theta$:

  (i)  draw random initial conditions and obtain 100 snapshots of the solution of the KS equation with parameter $\theta$;

  (ii)  interpolate snapshots onto $64 \times 64$ grid and compute sample covariance matrix;

  (iii)  compute largest eigenvalues of the sample covariance matrix and compute a log-linear fit.

The feature $\mathcal{M}_{\mathcal{F}}(\theta)$ consists of the slope and offset of the log-linear fit.

Finally, we need to chose a covariance matrix $R$. The perturbed observation approach (see section 3.1) is not useful here. If we assume that we collect data with measurement errors that are uncorrelated in space and time (adding an isotropic Gaussian to each snapshot), then this noise has no effect on the overall spatial scale of the solution and, thus, will not correctly reflect the uncertainty of the feature. The largest source of uncertainty in the feature is sampling error due the small number of snapshots

we use for computing the GP approximation. We can decrease the effects of this noise by using more snapshots, however this increases the computational cost. In addition, this uncertainty due to sampling error makes feature-based likelihood evaluations noisy, i.e., for a fixed $\theta$ and feature $f$, different runs of our simulations may lead to different likelihoods. This rules out Monte Carlo sampling for numerical solution of the data assimilation problem.

We address these issues by using a variational approach and compute an a posteriori estimate of $\theta$, i.e., we estimate $\theta$ by

maximizing the function

$$g(\theta) = \begin{cases} \exp\left(-\frac{1}{2}(f - \mathcal{M}_{\mathcal{F}}(\theta))' R^{-1} (f - \mathcal{M}_{\mathcal{F}}(\theta))\right) & \text{if } \theta \in [0, 5], \\ 0 & \text{otherwise}, \end{cases}$$

which is proportional to the feature-based posterior distribution. This will lead to a point-estimate for $\theta$ that leads to solutions that are compatible with the data. For point estimates, the covariance $R$ is not so essential. We set this covariance $R$ to be a diagonal matrix with diagonal entries $R_{11} = 2.25$, $R_{22} = 0.0625$. These values are chosen to reflect a relatively large amount

of uncertainty in the feature and to balance the different scales of the two components of the feature. However it is important to note that our approach does *not* allow us to draw conclusions about the uncertainty of our parameter estimate, for which we would require approximations of the posterior distribution. This may not be ideal, however in view of the computational difficulties, a point estimate is the best we can provide.

We need to decide on a numerical method for solving the optimization problem. Since the function $g$ is noisy and computa-

tionally expensive, we cannot compute its derivatives. Global Bayesian optimization (GBO, see, e.g., Frazier and Wang (2016)) is a derivative free method for optimization in exactly that setting, i.e., when the function to be optimized is computationally

expensive to evaluate and noisy. The basic idea of GBO is to model the function $g(\theta)$ by a Gaussian process (GP) and then to carefully chose additional points for evaluation of the function to improve the GP model. The maximizer of the mean of the GP model is then used to approximate the maximizer of the (random) function $g(\theta)$. We first explain how to build an initial GP model for the function $g(\theta)$ and then describe how to improve on the model given function evaluations. For more details about GBO, see Frazier and Wang (2016) or references therein.

A GP model for $g(\theta)$ consists of the mean and covariance functions

$$\mu(\theta) = \mu \quad \text{(constant mean function)},$$
$$C(\theta, \theta') = \sigma^2 \exp\left((\theta - \theta')^2 / L^2\right),$$

where $\mu, \sigma, L > 0$ are "hyperparameters" which we must define. To acknowledge the fact that $g(\theta)$ is noisy, we add another hyperparameter, $s > 0$, such that the covariance at the "observed points" $\theta_{\text{obs}}$ is given by $C(\theta_{\text{obs}}, \theta'_{\text{obs}}) + s$ (see section 3.3.5 of (Frazier and Wang, 2016)). We define the hyperparameters based on a small number of model function evaluations. Specifically, we evaluate $g$ at three points within $[0, 5]$ generated by a Sobol sequence, which is a space filling sequence of quasi-random points. This procedure suggests to evaluate the function at the boundaries and "in the middle" (see figure 11(a)). Given these three points $(\theta_i, g(\theta_i))$, $i = 1, 2, 3$, we maximize the "log marginal likelihood", which describes the probability of the three function evaluations $(\theta_i, g(\theta_i))$ (see section 3.3.6 of Frazier and Wang (2016)). This optimization is computationally inexpensive because it does not involve evaluating $g$ or solving the KS equation. We use an interior-point method (MATLAB's "fmincon") to carry out the optimization and enforce the bounds $0 \leq L \leq 1, 0.3 \leq \sigma^2 \leq 1, 0 \leq s \leq 0.1, 0 \leq \mu \leq 2$. This results in a crude approximation of $g$. We update this initial GP by the three function evaluations we already have, i.e., we recompute the mean $\mu$ and the covariance $C$, given these three function evaluations. The result is the GP illustrated in figure 11(a), where we show the mean (blue) and 200 samples (turquoise) of the updated GP, along with the three sample points (purple dots). Note that the GP model does not reflect the fact that $g(\theta)$ is non-negative. However, GBO is not easily modified to optimize non-negative functions.

To improve our GP model of $g(\theta)$ we wish to evaluate the function at additional points and we use the "expected improvement" criterion to determine these points. Expected improvement suggests points for additional evaluations of $g(\theta)$ using a trade-off between where the function is already known to be large and where the function is unknown (see section 3.4.1 of Frazier and Wang (2016)). This led to good results for our problem, however more advanced methods, e.g., knowledge gradient, may improve overall performance of the algorithm. We stopped the optimization when the integrated expected improvement is below a threshold ($10^{-4}$ in our case). With this set-up, we evaluated $g(\theta)$ five more times and computed the maximizer of $g(\theta)$ to be $\theta^* = 3.29$, which is near the parameter value we used to generate the feature data ($\theta_{\text{true}} = 3.38$).

The updated GP model is illustrated in the right panel of figure 11, where we show the mean (blue), the initial and additional points where $g(\theta)$ is evaluated (purple and red dots respectively) and 100 realizations of the updated GP model (turquoise). We also show 100 realizations of $g(\theta)$, obtained by evaluating $g(\theta)$ repeatedly over a grid of 100 equally spaced points. We note that the GP accurately describes the function and our confidence in the function for $\theta > 2.5$, where most of the function evaluations took place. The uncertainty is large for $\theta < 2.5$, which could be reduced by additional function evaluations. In

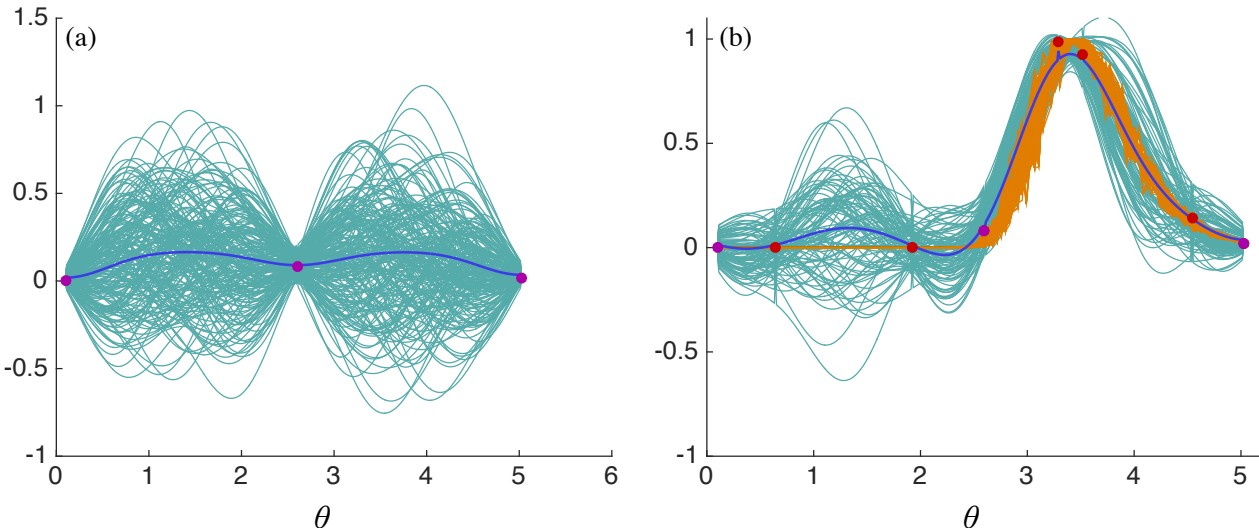

**Figure 11.** GP model of the function $g(\theta)$. (a) Initial GP model based on three function evaluations. Blue – mean function. Turquoise – 100 realizations of the GP. Purple dots – function evaluations. (b) Updated GP after GBO and 5 additional function evaluations. Blue – mean function. Turquoise – 100 realizations of the GP. Purple dots – initial function evaluations. Red dots – additional function evaluations based on expected improvement criterion. Orange – 100 samples of the random function $g(\theta)$.

summary, the feature-based approach, combined with an appropriate numerical technique for optimizing noisy functions, is successful in estimating a parameter of a chaotic partial differential equation.

## 5    Conclusions

We have discussed a feature-based approach to data assimilation. The basic idea is to compress the data into features and to
compute parameter estimates on posterior distributions defined in terms of the features, rather than the raw data. The feature-based approach has the advantage that one can calibrate numerical models to selected aspects of the data, which can help bridge gaps between low-dimensional models for complex processes and which can also help with breaking computational barriers in data assimilation with chaotic systems. Our main conclusions are as follows.

   (i) Constructing noise models directly for the features leads to straightforward numerical implementation of the feature-
based approach and enables the use of numerical methods familiar from data assimilation.

  (ii) The feature-based approach can reduce computational requirements by reducing an effective dimension. This reduction in complexity comes at the expense of a relaxation of how much that data constrain the parameters.

While the simplified noise models in (i) may lead to good results (in the sense that parameter estimates are useful) more work is needed to fully understand how to construct such noise models without excessive computations. Some of our numerical
examples indicate the limitations of the perturbed observations approach we propose for the construction of such noise models.

Our second conclusion (ii) suggests that one should use the feature-based approach only if the direct approach is infeasible. When the data can be compressed without loss of information, the feature-based approach is just as good or bad as the direct approach. The feature-based approach reduces computational requirements only if we truly reduce the dimension of the data by focussing only on *some* of the features of the data. In this case, one can formulate feature-based problems whose solution is straightforward, while a direct approach is hopeless.

*Code availability.* Code for the numerical examples will be made available on github: https://github.com/mattimorzfeld

*Competing interests.* No competing interests are present.

*Acknowledgements.* We thank Alexandre J. Chorin and John B. Bell from Lawrence Berkeley National Laboratory for interesting discussion and encouragement, and Bruce Buffett of the University of California at Berkeley for inspiration, encouragement and for providing his code for the B13 model. We thank Joceline Lega of the University of Arizona for providing code for numerical solution of the Kuramoto-Sivashinksy equation.

We thank three anonymous reviewers for insightful and careful comments which improved the manuscript.

MM, SL and RO gratefully acknowledge support by the National Science Foundation under grant DMS-1619630.

MM acknowledges support by the Office of Naval Research (grant number N00173-17-2-C003) and by the Alfred P. Sloan Foundation.

MM and JA were supported, in part, by National Security Technologies, LLC, under Contract No. DE-AC52-06NA25946 with the U.S. Department of Energy, National Nuclear Security Administration, Office of Defense Programs, and supported by the Site-Directed Research and Development Program. The United States Government retains and the publisher, by accepting the article for publication, acknowledges that the United States Government retains a non-exclusive, paid-up, irrevocable, worldwide license to publish or reproduce the published form of this manuscript, or allow others to do so, for United States Government purposes. The U.S. Department of Energy will provide public access to these results of federally sponsored research in accordance with the DOE Public Access Plan (http://energy.gov/downloads/doe-public-access-plan). The views expressed in the article do not necessarily represent the views of the U.S. Department of Energy or the United States Government. DOE/NV/25946-3357.

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
