# Peer review of "Feature-based data assimilation in geophysics"

_Nonlinear Processes in Geophysics, 2017_

## Referee Comment (RC1) · Anonymous Referee #1 · 30 Nov 2017

**GENERAL COMMENTS**

This paper is about parameter estimation problems in data assimilation. Conventional data assimilation techniques attempt to estimate parameters of a dynamical model by matching the model output to observations of the system. In some situations this is infeasible. The authors give the examples of a low-dimensional model of a complex system and a chaotic model over a long time-scale. In these situations the observations contain more information than can be matched. The authors show how (at least in some cases) this problem can be solved by extracting low-dimensional features from the observations and trying to match the model to these instead. A potential obstacle to this approach is the need to pass the observation noise model through a nonlinear mapping from observation space to feature space. The authors circumvent this obsta-

cle by the ingenious expedient of noting that observation noise models are often based on ad hoc additive Gaussian assumptions with little physical basis, and we are just as entitled to model the feature noise directly using such assumptions instead of trying to convert observation noise to feature noise.

The techniques described by the authors deserve to be better known. However, there are a number of ambiguities and minor errors in the exposition and experimental methodology. These should be corrected before publication in NPG.

SPECIFIC COMMENTS

p1, L24. The 'or vice versa' part of the sentence implies that data assimilation may be infeasible when the data have a lower dimension than the model. I don't understand how this could be. Indeed, this is one of the situations in which it has just been said (L22-23) that conventional data assimilation is not required.

p4, L2; p6, L27. When the likelihood is expressed in terms of the noise pdf, the noise pdf should be conditioned on \theta too. Thus the first case should be written l(z|\theta) = p_\epsilon(z-\mathcal{M}(\theta)|\theta).

p4, L4-5. I dispute that all types of prior information can be represented by a probability distribution. In particular, I disagree that knowledge of just lower or upper bounds (or both) can be so represented. However, I agree that in some cases prior information can be represented by a probability distribution, and in those cases the framework of the paper makes sense.

Following on from the previous point, the paper lacks justification for its choice of priors. This doesn't matter too much in the artificial examples (Examples 1 and 4), but in the examples with real data (Examples 2 and 3) the priors need to be justified if the results are to tell us anything about reality.

p4, L30-31. On my first reading I didn't understand how F(\theta) could be a random variable. It would help the reader to include an brief explanation of how this can arise.

p7, L7-8. R_f is not the sample covariance unless f is the sample mean.

SPECIFIC COMMENTS ON EXAMPLE 1

There is no statement about the initial conditions of the experiment. What were they?

p10. There is inconsistency over whether there is a data point at time 0. The formulae at L11 and L13 imply that there is not, as does L9 in referring to M as the number of data points. On the other hand, a data point at time 0 is shown in L12. On p11, Figure 1 shows a data point at time 0, but L1 of the caption implies that there is not one.

p10, L11. Is the variance of $v_i$ really 1? It looks much smaller in Figure 1.

p11, Figure 1, caption L1-2. The curves plotted are trajectories, not samples from the prior distribution of the parameters $p(\theta|z)$. This affects L4 too.

p11, L9. What is varied randomly in each experiment? The parameters? The initial conditions? Observation noise? Anything else?

p11, L9 says there were 100 experiments, but the caption of Figure 1 says there were 1000.

Figure 1 shows the mean of the KL divergence, but what about the spread? Conclusions such as p12, L2-3 would be ruined if the spread were much larger at M=100 than at M=500.

SPECIFIC COMMENTS ON EXAMPLE 2

As well as the aforementioned need to justify the prior, the choice of a unit covariance matrix in the observation noise (p13, L23) and the choice of covariance matrix in initialising the ensemble of walkers (p13, L30) need to be justified. Without these justifications it's not clear how (if at all) the results are connected to reality

SPECIFIC COMMENTS ON EXAMPLE 3

p15, L18 to p16, L1 and Figure 4. Is the MCD for B13 really the same as observed

over the past 30 Myrs? It looks much longer to me.

p16, L17. Are there really 150 values of MCD? If there are, the averaging windows must have been truncated at one end or the other. How was this done? It looks from Figure 5 that there are only 140 values (stopping 140 Myr ago).

p17, L4-5. What is the origin of t? It cannot be the present (as in Figure 5) if k in f_k is to be positive. Does \theta_k apply to the interval before or the interval after k.1Myr?

p17, L11. I assume that \theta_k is kept constant throughout the simulation. It would be useful to mention this here. What are the initial conditions of the simulation?

p17, L11-12. The part of the sentence after 'and' seems wrong. My understanding is that a single MCD is computed for the simulation, but this part implies that a sequence of values is calculated using a sliding window.

p18, L3-4. 100 simulations for each grid point or in total? (And if the latter, how were the simulations distributed about grid points?)

p18, Figure 6. Why is the \theta grid different for the two curves? What do the small circles on the orange curve represent?

p18, L15-16. The standard deviation should be a function of \theta_k, not of f_k. The relevant pdf of \eta_k is the one conditioned on \theta_k (see note on p4, L2; p6, L27). Substituting a standard deviation that is a function of f_k into the formula for a Gaussian pdf gives something that is the pdf of a Gaussian random variable when f_k is held fixed, but when \theta_k is held fixed it might not even be a valid pdf.

p18, L17-18; p19, L5-6. What are the justifications for these priors? They will need to be justified if the intention is to draw conclusions about physical reality.

SPECIFIC COMMENTS ON EXAMPLE 4

p20, L3-4. How is \theta determined for these integrations of the KS equation?

p20, L4. Over what range are the Fourier coefficients uniformly distributed?

p21, L1-2. How is \theta determined for these integrations of the KS equation?

p22, L17. Are these snapshots obtained in the same way as at p21, L1-4 or in some other way?

p23, L15. Even when it is restricted to be diagonal, R is not merely a scaling factor. The ratio of the diagonal elements determines which of the two components of the feature has to be matched most closely, and this can have a large effect on \theta.

p23, L16. As in my comments on Example 3, R should be a function of \theta, not of f.

p23, L20 onwards. Like, I suspect, most of the potential readership I'm unfamiliar with global Bayesian optimization and need a few more details to understand, even in outline, what is going on. I list specific points below. It would also be useful if the paper gave additional references for the method to increase the chance that at least one of them is in the library.

p24, L1. What is the log marginal likelihood and how does it allow us to estimate four hyperparameters from three function evaluations?

p24, L4-5. How is the updating of the GP done?

p24, L5. I assume that the \mu here is the mean function rather than the parameter. To avoid confusion it would be better to write \mu(\theta) here or adopt another notation for the parameter (such as \mu_0).

p24, L9. How were the extra function evaluations used to improve the GP model?

TECHNICAL CORRECTIONS

p4, L9. What is v? Is it a typo for \epsilon or something different?

p5, L14; p13, L24; p24, L3. 'Matlab' should be 'MATLAB' (at least that's how Math-Works spells it).

p5, L14; p13, L25. Grinsted lacks a year.

p5, (4). N_e hasn't been defined.

p5, L27. 'the the' should be 'the inverse'.

p5, L31 to p6, L1. 'm dimensional' should be hyphenated.

p6, L4. p_f(\theta) should be p_f(\theta|f).

p6, L11. 'are' should be 'is' (to agree with 'exception').

p6, L12. 'posterior distribution' should be 'likelihood'.

p6, L15. Insert comma after 'however'.

p6, L28. 'feature based' should be hyphenated.

p7, L4. Insert 'the' before 'EnKF'.

p7, L16; p8, L8. This is the third meaning of k. In Section 2.2 it was a sample index, and at the start of Section 3 (p5, L31) it was the dimension of data space. Could some other notation be found?

p7, L24. 'likelihoods' should be 'likelihood'.

p8, L8. The second 'values' should be 'vectors'.

p8, L18. Comma after 'see' should be before (and perhaps stronger).

p8, L20. Insert commas around 'however'.

p9, L3. Insert comma after 'hence'.

p9, L5. Insert comma before 'however'.

p9, L11. Hyphenate 'feature based'.

p9, L15. Delete comma after 'field'.

p9, L25-28. It would be useful to remind the reader which of the three types of section 3.3 example 4 is (as was done for the other three examples).

p10, L13. T should be t.

p10, L26. 'form' should be 'from'.

p10, L28. Period should be outside final parenthesis.

p11, Figure 1, top right. x-axis is duration, not number of data points.

p11, L1. 'suggest' should be 'suggests'.

p11, L2. 'were' should be 'we'.

p11, L4. Delete second 'posterior distributions'.

p11, L4. $p_{450}$ should be $p_{500}$ and not followed by a comma.

p11, (8). Drop subscripts on $z_M$ for consistency with notation on p10.

p12, L9. 'features' should be 'feature'.

p12, L10-11. This sentence lacks a main verb.

p12, L10-14. For consistency with the notation of Section 3.1, R should be R_f.

p12, L14-15. For consistency with the notation of Section 3.1, \mathcal{F}_\mathcal{M} should be mathcal{M}_\mathcal{F}.

p13, L15-17. The curious reader will appreciate being told at this point that the data can be seen in Figure 3.

p13, L20-21. On first reading the first sentence I couldn't understand how singular values and vectors had been got from the data. By the end of the second sentence it was clear that the data had been arranged in a 2x11 matrix. It would be good to mention this first.

p13, L24. 'feature based' should be hyphenated.

p14, Figure 2. The notation used for the axis labels is a mixture of subscripted \theta-components and parameter names. It would be neater to change the first four labels to \alpha, \beta, \gamma, \delta.

p15, L12. \theta(t) should not be here. It hasn't been introduced yet.

p15, L18; p18, L7; p23, L7, L18; p24, L12. Stronger punctuation is needed before 'however' (and perhaps a comma afterwards). With just a comma before 'however' I was expecting a sentence of the form 'The EnKF will not work with these data, however much the covariance in inflated'. Instead, what we have is 'The EnKF will not work with these data; however, the WEnKF will'.

p16, L4. Delete space before comma.

p18, Figure 6, caption, L2. 'bards' should be 'bands'.

p18, L4. 'reversal' should be 'reversals'.

p18, L8. 'upper' and 'lower' should be swapped for consistency with the rest of the sentence.

p19, L8-9. 'right' should be 'left' and vice versa.

p19, L28. The domain in Figure 9 is [0,10]\times[0,10].

p20, L2. Delete first 'different'.

p22, L2. 'features' should be 'feature'.

p22, L13. 'feature based' should be hyphenated.

p23, L6. Insert 'to' after 'due'.

p23, L11. 'issue' should be 'issues'.

p23, L30. Based on the table of contents of Frazier and Wang, it looks like 'chapter 3.5'

should be 'section 3.3.5'.

p24, L12. 'lead' should be 'led' or 'leads'.

p24, L15. 'generated' should be 'generate'.

p27, L18. Volume and page numbers should be '198, 597-608'.

p28, L12, L14. The title of the same journal is given in two different ways.

p28, L26. This is a chapter in a monograph but has been formatted as though it were a paper in a journal.

p29, L4. The second 'reversals' should be 'field'.

p29, L7. Unlike the titles of other journal papers, this one has been capitalised.

p29, L12. '4144' should be '4114'.

---

## Referee Comment (RC2) · Anonymous Referee #2 · 20 Dec 2017

Manuscript ID: npg-2017-52

General Comments:

This study evaluates the concept of "feature-based data assimilation (DA)" in four simple geophysical models by employing three numerical techniques: Markov Chan Monte Carlo, direct sampling, and Global Bayesian Optimization. The true test would come when evaluating "feature based DA" in more-complex geophysical modeling systems. Nonetheless, the results presented offer encouragement, especially because the basic ideas presented in this article can be applicable to some of the machine learning algorithms used in data assimilation. The topic of the manuscript is within the scope of this journal. However, some of the concepts presented require more detailed explanations, which need to be addressed before final publication. Specific details are provided in

the following sections.

Specific Comments:

1. Writing style needs to be improved in some parts of the manuscript. The phrase "model output and data" or the words "model" and "data" are repeatedly used throughout the text. This makes sense, since they are the central themes of this paper. However, some sentences could be rephrased to avoid repetition and thus continue to engage the reader. Only some of the typos and grammatical deficiencies encountered were included in the "Technical Corrections" section.

2. P1, L5: Clarify this statement "assimilate data of a complex-system into a lower-dimensional model." In data assimilation problems, the most common scenario is to have a model state with dimensions much larger than those of an observation vector. Can you provide a few examples in the text rather than pointing the reader directly to some references.

3. Please, elaborate on the applicability of "feature-based DA" when assimilating data into higher-dimensional models (e.g. coupled atmosphere-chemistry or coupled atmosphere-ocean numerical models)? See the work by Yablonski and Ginis, 2007 for reference.

4. P9, L10: Explain what is meant by "different scales" are you referring to time or space?

5. P13, L34: Provide an explanation to the strong correlation between the parameters, but not so much to the initial conditions. Also, did you evaluate correlations between pairs of parameters?

6. P14, L4-9: Further clarify the differences on the trajectories for the Here and Lynx populations. Is the model evaluated at each cycle-step, whereas the data are cumulative? Whys is this a limitation? In general data assimilation problems, better fits to the observations are found by assimilating continuous data and not just data at a
given cycle. What limitations in LV model would lead to such different behavior on the trajectories of those critters.

7. P16, L16-19: What are the advantages of using an "averaging window"? Is it possible that trying to find an optimal multiplicative parameter ïĄś(t) through minimization of a variational cost function would produce better results? Did you consider this approach as an alternative to particle filters?

Technical Corrections:

1. Labeling figures using letters e.g. (a), (b), etc. will ease finding their reference in the text.

2. P1, L6: Delete the comma before "and." Two nouns that appear There are many instances of this throughout the text Don't separate two nouns that appear together as a compound subject or compound object.

3. P3, L23: Choose "and" or "but"

4. P4, L10: Replace "all over" with "in"

5. P4, L19: Choose either "as in the case" or "for example"

6. P5, L24: Replace "one thus wants" with "one thus want"

7. P5, L25: Same as in 2

8. P9, L23: The abbreviation for a "million years" is myr or Myr, correct throughout the text

9. P9, L30: Same as in 2

10. P10, L1: Insert a comma before "but"

11. P10, L26: Replace "form" with "from"

12. P11, L5: "posterior distributions" is repeated twice

13. P13, L22: Insert "of" in between "mean" and "zero"

14. P13, L26: Replace the period next to "as follow" with a colon

15. P13, L30: Move the comma inside the quotation marks

16. P15, L13: What does R stands for?

17. P16, L13: Use the past participle form of to vary

18. P21, L1: Replace the period after "follows" with a comma

19. P21, L22: Delete the comma after "process"

20. P21, L25: Same as in 18

21. P22, L5: replace the colon after "terms" with a comma

22. P23, L7: missing period

23. P23, L11: Use plural form of "issue" and revise comma use

24. P24, l12: Use "leads" or "led"

25. P25, L8: Revise the use of the word "base"

Figures:

1. Figure 1: label figures (a) – (d) and insert those into the body of the manuscript for easier reading, repeat for all Figures with multiple plots

References:

Yablonsky, R. M., and I. Ginis 2008: Improving the Ocean Initialization of Coupled Hurricane–Ocean Models Using Feature-Based Data Assimilation, Monthly Weather Review, 136, 2592–2607.

---

## Referee Comment (RC3) · Anonymous Referee #3 · 22 Dec 2017

The manuscript presents a novel approach to data assimilation in which the assimilation algorithm is applied not to the raw data but to 'features' extracted from the data. A key insight is that one can readily construct these algorithms by devising a noise model for the features, allowing the use of standard DA algorithms.

The article makes a worthy contribution to the literature and in general I am happy with the presentation and content. I have one 'big picture' criticism which is that the problem of estimating the likelihood for features, and the solution presented in the manuscript, are not discussed except in quite mundane terms. I would like to see a few more sentences of interpretation, intuition or speculation introduced on the feature-based noise model, for instance discussing the Gaussian assumption made in the perturbed observations approach, or the nonGaussian precursor to the feature likelihood (5).

I think for the former case that the Gaussian assumption is mostly a modelling construct – but some intuition suggests that if the feature is appropriate to the parameters and the parameters are not redundant in their effect, then the likelihood function is probably unimodal, and in this case the Gaussian assumption might not be so bad. Is this intuition correct?

For evaluating (5), I wonder if a particle-based approach could represent the likelihood in more fidelity. In this case, also using a particle-based approach for the prior, would one require an ensemble of particles to represent the likelihood for each particle in the prior? Can this possibly be constructed as a Rao-Blackwellized particle filter?

Everything I am asking for is speculative – because I feel some speculation is called for. Let me emphasise that I do not necessarily want each of the above questions to be answered definitively – that would use a great deal of space. I would, however, like to hear more of the authors' opinions on the viability of alternate approaches, and realistic appraisals of the current approach.

P2, line 8-11: From P1 "The likelihood connects the model and its parameters to the data, and is often based on the mismatch of model output and data. A typical example is the squared two-norm of the difference of model output and data."

I would then expect that feature-based data assimilation would entail extracting features from both, model output and data. I think it is clear that this is done in section 4.1. But lines 8-11 on P2 make it sound like one only extracts features from the data.

P7: the noise model for the features is constructed by studying perturbations of the observations in a manner similar to the EnKF. It might be useful to note that the number of perturbed data $N\_z$ will be limited by computational power to depend on the number of observations and the complexity of the feature extraction algorithm, and that as in the EnKF the rank of the extracted covariance matrix will depend on $N\_z$.

P7, line 16: a comment on the (non)Gaussianity of the distribution of singular vectors

would be welcome, if anything is known, given that SVD is a common method for extracting features or summary statistics.

I think some extra care needs to be taken when defining the cases and effective dimension on p8. These are referred to extensively on p9 and I think the discussion is unnecessarily hard to follow.

P8: Case (ii) is simply "several aspects of the data are neglected" but what is really meant (I think) is that several important or information carrying aspects of the data are neglected, in order to contrast with case (i). I will proceed assuming this is the case; even if I am incorrect, I think a later paragraph (P9, line 11, see below) will need careful rephrasing.

P8: line 39 defines an 'effective' dimension, but line 33 appears to me to refer to the same object as 'intrinsic' dimension, which I see is the term used in the reference Agapiou et al. . .

P8, final line: "Intuitively, the more information the data contains about the parameters, the harder is the problem."

This sentence is very confusing on the first several readings. Suppose the data consist of perfect recordings of the model parameters – then the DA problem is easy. I suppose this is a way to describe situations in which e.g. the likelihood is narrow and/or has little intersection with the prior, among many other pathological scenarios for DA schemes – but it is worth remembering that for someone unfamiliar with DA, the idea that problems arise from having very informative data is not intuitive at all but instead extremely unintuitive.

P9, line 11: "In case (ii) however, the feature changes the posterior distribution and, hence, the effective dimension. Since the feature neglects several aspects of the data, assimilating the feature will introduce a more gradual change from prior to posterior distribution than if all data are used. Thus, the feature-based approach reduces the

effective dimension of the problem. For chaotic systems, this reduction in effective dimension can be so dramatic that the original problem is infeasible, while a feature-based approach becomes feasible, see Hakkarainen et al. (2012); Haario et al. 10 (2015); Maclean et al. (2017) and section 4.4."

I think the middle sentence must be incorrect, and I am sceptical about the third. If assimilating the feature always leads to a more gradual progression from prior to posterior (from the second sentence), and if this is the main source of the reduction in effective dimension, then how could feature DA be feasible in scenarios when the original problem is infeasible (from the third)? I think what is missing is that the computational cost of these schemes falls if the feature is lower dimensional than the data.

The loose definition of case(ii) trips us up again here – it should be made clear that some useful data is being thrown out. Second, I find that case (ii) is not established clearly enough later on in Sec 4.4. Some more care should be taken to discuss whether information, and what information, is being lost or not in the numerical example considered. I bring this point up again later but it is crucial for the discussion here that it be addressed.

P13: While examples 1,3,4 are well justified, there is not really a need to use features to resolve the LV DA problem in 4.2. This is mentioned in for instance the conclusion, but I would appreciate a motivating note at the opening of 4.2, informing the reader that this is a demonstration of the feature method in case (i).

P20-21: I suggest figure 8 come before figure 7, and be referred to on line 7 on p19 as 'our results'. Figure 8 clearly shows the improvement in the models over the poor initial representations in figure 4, while figure 7 is an evaluation in feature space. If at all possible, the horizontal axis of figure 4 should be extended out to -150 Myr to match the scale of figure 8.

P22: The feature is well chosen to capture the influence of the parameter. What information is lost by choosing to use the feature? Can one show that the feature chosen is

not a sufficient statistic, even by a heuristic argument? Again this comment harks back to my old complaints about case (i) vs case (ii)...

P25: "The feature-based approach reduces computational requirements only if we truly reduce the dimension of the data by focussing only on some of the features of the data."

I think this is a useful comment and in some ways the focus on effective dimension is unfortunate because it obscures this component of the discussion. I wish a comment like this appeared more clearly back on P8.

---

## Author Comment (AC1) · 27 Jan 2018

**Feature-based data assimilation in geophysics – response to reviewers**

Matthias Morzfeld, Jesse Adams, Spencer Lunderman, and Rafael Orozco

We thank all three reviewers for their careful and insightful comments. Our manuscript was greatly improved by incorporating your comments. You can find detailed responses to each of your comments below. We also attach a document that contains all changes we made during this revision process. We find this version unreadable and also attach the revised manuscript without highlighting the changes we made.

**Response to Reviewer 1**

**Specific comments**

1. *p1, L24. The "or vice versa" part of the sentence implies that data assimilation may be infeasible when the data have a lower dimension than the model. I don?t understand how this could be. Indeed, this is one of the situations in which it has just been said (L22-23) that conventional data assimilation is not required.*

   You are right, we removed the "vice versa" part. It is possible (in principle) that a model be higher-dimensional than data, but we do not discuss this case here (see also comment 3 of Reviewer 2).

2. *p4, L2; p6, L27. When the likelihood is expressed in terms of the noise pdf, the noise pdf should be conditioned on $\theta$ too. Thus the first case should be written $l(z|\theta) = p_\epsilon(z - \mathcal{M}(\theta)|\theta)$.*

   You are right, we corrected this error.

3. *p4, L4-5. I dispute that all types of prior information can be represented by a probability distribution. In particular, I disagree that knowledge of just lower or upper bounds (or both) can be so represented. However, I agree that in some cases prior information can be represented by a probability distribution, and in those cases the framework of the paper makes sense.*

   *Following on from the previous point, the paper lacks justification for its choice of priors. This doesn't matter too much in the artificial examples (Examples 1 and 4), but in the examples with real data (Examples 2 and 3) the priors need to be justified if the results are to tell us anything about reality.*

   You are right, our claim is too strong and we softened our statement. Bounds on parameters can be incorporated by assuming uniform distributions over parameters.

   We also revised the manuscript to emphasize the importance of priors. Throughout the examples with "real data", we use what we believe are appropriate priors. For example 2, we have essentially no prior information about the parameters and hence chose a uniform prior (we made this more explicit in our revision). For example 3, the "first" prior (150 Myr ago) is quickly forgotten and all subsequent priors in this sequential problem are based on results from the previous data assimilation(s). This is typical in particle filtering.

4. *p4, L30-31. On my first reading I didn't understand how $F(\theta)$ could be a random variable. It would help the reader to include an brief explanation of how this can arise.*

   We revised this part of the manuscript, extended our description and present examples of "random" $F$'s.

5. *p7, L7-8. $R_f$ is not the sample covariance unless $f$ is the sample mean.*

   You are right, we corrected this error.

**Specific comments on example 1**

1. *There is no statement about the initial conditions of the experiment. What were they?*

   The initial conditions are $x(0) = 0$, $dx/dt = 0$. We included the initial conditions in the revised manuscript.

2. *p10. There is inconsistency over whether there is a data point at time 0. The formulae at L11 and L13 imply that there is not, as does L9 in referring to M as the number of data points. On the other hand, a data point at time 0 is shown in L12. On p11, Figure 1 shows a data point at time 0, but L1 of the caption implies that there is not one.*

   There is a data point at time $t = 0$. We corrected errors arising from us being imprecise throughout this section.

3. *p10, L11. Is the variance of $v_i$ really 1? It looks much smaller in Figure 1.*

   You are right, this is a typo. The variance is $r = 0.001$. We checked our code and the code is consistent.

4. *p11, Figure 1, caption L1-2. The curves plotted are trajectories, not samples from the prior distribution of the parameters $p(\theta|z)$. This affects L4 too.*

   You are right, we corrected this error.

5. *p11, L9. What is varied randomly in each experiment? The parameters? The initial conditions? Observation noise? Anything else?*

   Only the observation noise was varied in each experiment. We clarified this in our revision.

6. *p11, L9 says there were 100 experiments, but the caption of Figure 1 says there were 1000.*

   We did 1000 experiments and corrected the typo in our revision.

7. *Figure 1 shows the mean of the KL divergence, but what about the spread? Conclusions such as p12, L2-3 would be ruined if the spread were much larger at M=100 than at M=500.*

   We included the spread (two standard deviation error bars) in our revision. We computed a "small" spread and thus are reassured that our conclusions are valid.

**Specific comments on example 2**

*As well as the aforementioned need to justify the prior, the choice of a unit covariance matrix in the observation noise (p13, L23) and the choice of covariance matrix in initialising the ensemble of walkers (p13, L30) need to be justified. Without these justifications it's not clear how (if at all) the results are connected to reality*

We clarified our choice of prior. We also clarified that the choice of the initial ensemble is not crucial if we generate a Markov Chain that is "long enough". We experimented with several initial ensembles for our ensemble sampler and found that the choice of initial ensemble is not crucial. This is in fact one of the strengths of the affine invariant ensemble method.

**Specific comments on example 3**

1. *p15, L18 to p16, L1 and Figure 4. Is the MCD for B13 really the same as observed over the past 30 Myrs? It looks much longer to me.*

   No, MCD for B13 is not the same as observed over the past 30 Myr. We state that the MCD is "comparable". The MCDs being on the "same order of magnitude" however is more accurate and we use this formulation in our revision.

2. *p16, L17. Are there really 150 values of MCD? If there are, the averaging windows must have been truncated at one end or the other. How was this done? It looks from Figure 5 that there are only 140 values (stopping 140 Myr ago).*

   You are right, our presentation is sloppy here. There are 157.5 Myr of data, averaging (always from past to present) over 10 Myr leads to 149 MCD. We clarified this in the revision and also changed figure captions and labels. Thank you for pointing this out and for the careful counting.

3. *p17, L4-5. What is the origin of t? It cannot be the present (as in Figure 5) if k in $f_k$ is to be positive. Does $\theta_k$ apply to the interval before or the interval after k.1Myr?*

   You are right, our presentation is sloppy here. We revised our indexing and expanded the explanations.

4. *p17, L11. I assume that $\theta_k$ is kept constant throughout the simulation. It would be useful to mention this here. What are the initial conditions of the simulation?*

   Yes, $\theta$ is constant during these simulations and we stated this explicitly in our revision. The initial condition for all simulations is $x(0) = 0$ and we state that in our revision as well.

5. *p17, L11-12. The part of the sentence after "and" seems wrong. My understanding is that a single MCD is computed for the simulation, but this part implies that a sequence of values is calculated using a sliding window.*

   You are right, our presentation was confusing here. We clarified this issue in our revision.

6. *p18, L3-4. 100 simulations for each grid point or in total? (And if the latter, how were the simulations distributed about grid points?)*

   We did 100 simulations per grid point.

7. *p18, Figure 6. Why is the $\theta$ grid different for the two curves? What do the small circles on the orange curve represent?*

   The $\theta$ grid is chosen to cover (roughly) the same values of corresponding MCDs (or features $f$). We chose a slightly finer grid for the P09 model because simulations with this model are quicker.

8. *p18, L15-16. The standard deviation should be a function of $\theta_k$, not of $f_k$. The relevant pdf of $\eta_k$ is the one conditioned on $\theta_k$ (see note on p4, L2; p6, L27). Substituting a standard deviation that is a function of $f_k$ into the formula for a Gaussian pdf gives something that is the pdf of a Gaussian random variable when $f_k$ is held fixed, but when $\theta_k$ is held fixed it might not even be a valid pdf.*

   We disagree. It is common to adjust error variances based on how large a measured quantity is and this is the approach we use here. Note that $f$ is fixed in a posterior distribution.

9. *p18, L17-18; p19, L5-6. What are the justifications for these priors? They will need to be justified if the intention is to draw conclusions about physical reality.*

You are right, variances in our priors are not sufficiently justified. The prior variance at the first step however is quickly "forgotten" in our sequential set up. The prior variance at all subsequent steps is chosen to have a mildly regularizing effect. We considered several values for these variances and did not observe major differences in our results. Once we use our approach to draw conclusions about the history of Earth's core, we will revise and refine our choices. In this paper, we want to demonstrate how to use the feature-based approach and show that it can be successful in real problems with real data. We believe that our choices of priors are reasonable for these purposes.

**Specific comments on example 4**

1. *p20, L3-4. How is $\theta$ determined for these integrations of the KS equation?*

We describe our simulation/likelihood procedure for a given $\theta$. We revised the manuscript and explained this in more detail.

2. *p20, L4. Over what range are the Fourier coefficients uniformly distributed?*

Each Fourier coefficient is drawn from a uniform distribution on $[0, 1]$.

3. *p21, L1-2. How is $\theta$ determined for these integrations of the KS equation?*

As above, we describe our simulation/likelihood procedure for a given $\theta$. We revised the manuscript and explained this in more detail.

4. *p22, L17. Are these snapshots obtained in the same way as at p21, L1-4 or in some other way?*

These snapshots are obtained in the same way. We added a clarification in our revision.

5. *p23, L15. Even when it is restricted to be diagonal, R is not merely a scaling factor. The ratio of the diagonal elements determines which of the two components of the feature has to be matched most closely, and this can have a large effect on $\theta$.*

You are right. We explained our choice of $R$ in more detail in our revision.

6. *p23, L16. As in my comments on Example 3, R should be a function of $\theta$, not of f.*

$R$ is actually a constant as stated in the revised manuscript.

7. *p23, L20 onwards. Like, I suspect, most of the potential readership I'm unfamiliar with global Bayesian optimization and need a few more details to understand, even in outline, what is going on. I list specific points below. It would also be useful if the paper gave additional references for the method to increase the chance that at least one of them is in the library.*

We revised the manuscript to provide a bit more detail about Bayesian optimization. We are unable to provide full detail due to length restrictions (and because our paper is not about Bayesian optimization). We agree with you that this is unsatisfactory, however we use a variety of numerical methods and none of them are explained in detail. Our point is that our (conceptual) approach is flexible and can be used with a variety of methods. The paper is understandable when thinking of some numerical methods as "black box algorithms" that can be used to solve a given problem. If a reader is unfamiliar with a particular method, e.g., global optimization, affine invariance ensemble MCMC, but wants to learn more, then

we have to use references or else our paper will be long and unreadable. The reference we cite for Bayesian optimization is in the arxiv (arxiv.org) and free to access for everyone: https://arxiv.org/abs/1506.01349 No need to visit any library.

**Technical corrections**

We corrected all "technical corrections" (typos, grammar issues, repeated words etc.). We also revised our use of $k$ in three ways. Thank you for bringing all these issues to our attention.

**Response to Reviewer 2**

**Specific comments**

1. *Writing style needs to be improved in some parts of the manuscript. The phrase "model output and data" or the words "model" and "data" are repeatedly used through- out the text. This makes sense, since they are the central themes of this paper. However, some sentences could be rephrased to avoid repetition and thus continue to engage the reader. Only some of the typos and grammatical deficiencies encountered were included in the "Technical Corrections" section.*

   Thank you for helping us improve our writing style. We made all technical corrections you asked for and also made an effort to improve our writing abilities and revise the manuscript. We feel that writing style is less important than documenting relevant scientific/mathematical results.

2. *P1, L5: Clarify this statement "assimilate data of a complex-system into a lower-dimensional model." In data assimilation problems, the most common scenario is to have a model state with dimensions much larger than those of an observation vector. Can you provide a few examples in the text rather than pointing the reader directly to some references.*

   We are a bit puzzled by this comment. Line 5 is in the abstract, we do not provide references. We believe you are addressing p1. L25. We include specific examples in our revision and agree with you that this is more to the point than just providing references.

3. *Please, elaborate on the applicability of "feature-based DA" when assimilating data into higher-dimensional models (e.g. coupled atmosphere-chemistry or coupled atmosphere-ocean numerical models)? See the work by Yablonski and Ginis, 2007 for reference.*

   We exclude this case in our paper and focus on the three scenarios we describe in section 3.3. However, you are right that this scenario (assimilating data into models that are higher-dimensional than the data) is important and we mention this in our revision. We also cite the suggested reference, which is relevant and which we were unaware of. Thank you for bringing this work to our attention.

4. *P9, L10: Explain what is meant by "different scales" are you referring to time or space?*

   Both scales, spatial and temporal, could be different. We clarify this in our revision.

5. *P13, L34: Provide an explanation to the strong correlation between the parameters, but not so much to the initial conditions. Also, did you evaluate correlations between pairs of parameters?*

   We do not have an explanation for the weak correlation between parameters and initial conditions. This is something that the algorithm discovered for us. For the purposes of this paper, we are not so much interested in the implications of our results, but in producing results with our proposed numerical and computational framework. To be sure, we obtain very similar strong/weak correlations with the classical approach and this is the main point of this example. We made an effort to clarify all this in our revision.

6. *P14, L4-9: Further clarify the differences on the trajectories for the Here and Lynx populations. Is the model evaluated at each cycle-step, whereas the data are cumulative? Why is this a limitation? In general data assimilation problems, better fits to the observations are*

*found by assimilating continuous data and not just data at a given cycle. What limitations in LV model would lead to such different behavior on the trajectories of those critters.*

We apply data assimilation only to a small part of the Lynx-Hare data set because the data oscillates with time-varying amplitudes. The Lotka-Volterra (LV) equations are incapable of such oscillations (LV amplitude is fixed). It is known that the LV model has limitations and the fixed amplitude of oscillations is one of these limitations. As stated above, and as emphasized in the revision, we use the LV model and the Lynx-Hare data merely as a demonstration of the robustness of our approach. In fact, this problem is simple enough to solve with the "standard" data assimilation approach and we emphasize this point as well. This issue was also brought up by Reviewer 1.

7. *P16, L16-19: What are the advantages of using an "averaging window"? Is it possible that trying to find an optimal multiplicative parameter through minimization of a variational cost function would produce better results? Did you consider this approach as an alternative to particle filters?*

8. The advantage of a feature (in this example, an averaging window) is that the feature-based approach makes the problem easier to solve. The "raw data" are the sign of the variable of the SDE. Noise modeling is hard for this type of observation (what is the meaning of "noise" added to a "plus sign"?). We have not tried a variational approach in part because the cost-function is not easy to derive for this observation (the sign of a quantity).

**Technical corrections**

We corrected all "technical corrections" (typos, grammar issues, repeated words etc.). We have also revised how we label our figure and sub-figures. Thank you for bringing all these issues to our attention.

**Response to Reviewer 3**

**Specific comments**

1. *I would like to see a few more sentences of interpretation, intuition or speculation introduced on the feature-based noise model, for instance discussing the Gaussian assumption made in the perturbed observations approach, or the nonGaussian precursor to the feature likelihood (5). I think for the former case that the Gaussian assumption is mostly a modeling construct ? but some intuition suggests that if the feature is appropriate to the parameters and the parameters are not redundant in their effect, then the likelihood function is probably unimodal, and in this case the Gaussian assumption might not be so bad. Is this intuition correct?*

   Yes, you are correct. We added a clarification of these issues in our revision. Thank you for bringing this to our attention.

2. *For evaluating (5), I wonder if a particle-based approach could represent the likelihood in more fidelity. In this case, also using a particle-based approach for the prior, would one require an ensemble of particles to represent the likelihood for each particle in the prior? Can this possibly be constructed as a Rao-Blackwellized particle filter?*

   These are very interesting ideas which we can consider in the future (perhaps with your help). We feel that a detailed investigation of how to evaluate (5) is out of the scope of this paper (which advocates for replacing (5) by another equation).

3. *Everything I am asking for is speculative ? because I feel some speculation is called for. Let me emphasize that I do not necessarily want each of the above questions to be answered definitively ? that would use a great deal of space. I would, however, like to hear more of the authors? opinions on the viability of alternate approaches, and realistic appraisals of the current approach.*

   Some of us spent a lot of time on studying limitations of particle filters (PF). While we have not investigated how feasible PFs are for evaluating/modeling equation (5), we are aware of several computational difficulties that can arise when applying PFs in high-dimensional settings. In such cases, building a PF for (5) can become a major computational task. It is difficult to anticipate (speculate) about whether or not this is feasible in some cases, or if it can be competitive with our proposed feature-based approach. We believe that making general statements is out of reach because PFs (and their limitations) and the feature-based approach are not understood well enough. We feel that speculating on some specific examples is not a good idea for a team of young scientists like us and, for that reason, hesitate to speculate in the manuscript. We believe that our four examples allow for some speculation, but this speculation should be done by the reader. We will certainly test our approach more often and plan to report on future success (and failures) in the near future.

4. *P2, line 8-11: From P1 "The likelihood connects the model and its parameters to the data, and is often based on the mismatch of model output and data. A typical example is the squared two-norm of the difference of model output and data." I would then expect that feature-based data assimilation would entail extracting features from both, model output and data. I think it is clear that this is done in section 4.1. But lines 8-11 on P2 make it sound like one only extracts features from the data.*

   You are right, our presentation was imprecise here. We addressed your comment and clarified in our revision that the model also produces the feature.

5. *P7: the noise model for the features is constructed by studying perturbations of the observations in a manner similar to the EnKF. It might be useful to note that the number of perturbed data $N_z$ will be limited by computational power to depend on the number of observations and the complexity of the feature extraction algorithm, and that as in the EnKF the rank of the extracted covariance matrix will depend on $N_z$.*

   You are absolutely right. We included a discussion about the rank of the covariance and possible computational limitations in our revision.

6. *P7, line 16: a comment on the (non)Gaussianity of the distribution of singular vectors would be welcome, if anything is known, given that SVD is a common method for extracting features or summary statistics.*

   We are unaware of any general results about distributions of singular vectors.

7. *I think some extra care needs to be taken when defining the cases and effective dimension on p8. These are referred to extensively on p9 and I think the discussion is unnecessarily hard to follow. P8: Case (ii) is simply ?several aspects of the data are neglected? but what is really meant (I think) is that several important or information carrying aspects of the data are neglected, in order to contrast with case (i). I will proceed assuming this is the case; even if I am incorrect, I think a later paragraph (P9, line 11, see below) will need careful rephrasing.*

   You are correct – the main point of case (ii) is to actually neglect data, not to "just" represent it more effectively (case (i)). This comes up in several of your comments. We revised the whole section to properly address your comment and hope that our revised presentation is more clear. We also preview one of our main conclusions, i.e., that the feature-based approach is computationally advantageous only if some aspects of the data are neglected.

8. *P8: line 39 defines an "effective" dimension, but line 33 appears to me to refer to the same object as "intrinsic" dimension, which I see is the term used in the reference Agapiou et al.*

   Definitions and concepts of effective/intrinsic dimensions are not well established in the current literature. There are several effective dimensions and some of them are called "effective dimension", some "intrinsic dimension" (see Agapiou et al., section 3.2 and Chorin and Morzfeld 2013). We mean "effective dimension" throughout the manuscript and fixed this issue in the manuscript.

9. *P8, final line: "Intuitively, the more information the data contains about the parameters, the harder is the problem." This sentence is very confusing on the first several readings. Suppose the data consist of perfect recordings of the model parameters – then the DA problem is easy. I sup- pose this is a way to describe situations in which e.g. the likelihood is narrow and/or has little intersection with the prior, among many other pathological scenarios for DA schemes – but it is worth remembering that for someone unfamiliar with DA, the idea that problems arise from having very informative data is not intuitive at all but instead extremely unintuitive.*

   You are right, what is intuitive for us is not intuitive for everybody. We are used to thinking about effective dimension and using the definition of effective dimension in section 3.3 (see above comment and response) the statement "the more information the data contains about the parameters, the harder is the problem" is intuitive. We made an effort to revise the manuscript to explain this more carefully so that this section is easier to follow.

10. *P9, line 11: "In case (ii) however, the feature changes the posterior distribution and, hence, the effective dimension. Since the feature neglects several aspects of the data, assimilating the*

*feature will introduce a more gradual change from prior to posterior distribution than if all data are used. Thus, the feature-based approach reduces the effective dimension of the problem. For chaotic systems, this reduction in effective dimension can be so dramatic that the original problem is infeasible, while a feature-based approach becomes feasible, see Hakkarainen et al. (2012); Haario et al. 10 (2015); Maclean et al. (2017) and section 4.4."*

*I think the middle sentence must be incorrect, and I am skeptical about the third. If assimilating the feature always leads to a more gradual progression from prior to posterior (from the second sentence), and if this is the main source of the reduction in effective dimension, then how could feature DA be feasible in scenarios when the original problem is infeasible (from the third)? I think what is missing is that the computational cost of these schemes falls if the feature is lower dimensional than the data. The loose definition of case(ii) trips us up again here – it should be made clear that some useful data is being thrown out. Second, I find that case (ii) is not established clearly enough later on in Sec 4.4. Some more care should be taken to discuss whether information, and what information, is being lost or not in the numerical example considered. I bring this point up again later but it is crucial for the discussion here that it be addressed.*

You are right. When dimensionality of the data is low, the computational requirements are also low. This is what "effective dimension" tells us. We have already responded to criticism of the lose definition of case (ii) above and we hope that all these important points are easier to understand in our revised version. We completely re-wrote and re-structured section 3.3. Thank you for helping us making our points more clearly and more concisely.

11. *P13: While examples 1,3,4 are well justified, there is not really a need to use features to resolve the LV DA problem in 4.2. This is mentioned in for instance the conclusion, but I would appreciate a motivating note at the opening of 4.2, informing the reader that this is a demonstration of the feature method in case (i).*

You are right, we clarified this in our revision and added a sentence that highlights that this example is for demonstration purposes only, "classical DA" is not out of reach here.

12. *P20-21: I suggest figure 8 come before figure 7, and be referred to on line 7 on p19 as "our results". Figure 8 clearly shows the improvement in the models over the poor initial representations in figure 4, while figure 7 is an evaluation in feature space. If at all possible, the horizontal axis of figure 4 should be extended out to -150 Myr to match the scale of figure 8.*

We do not feel strongly about this, but prefer the order we have. "Our results" really are in feature space. Figure 8 is only indicative of what the feature-space results provide. It is one stochastic simulation, other simulations, using different random number generator seeds, will lead to different trajectories (but all trajectories have similar MCD variation). The issue with the time axis was also brought up by Reviewer 1 and we fixed these issues.

13. *P22: The feature is well chosen to capture the influence of the parameter. What information is lost by choosing to use the feature? Can one show that the feature chosen is not a sufficient statistic, even by a heuristic argument? Again this comment harks back to my old complaints about case (i) vs case (ii).*

The information that is "lost" is that the feature-based approach does not yield trajectories that match the data point-wise. A point-wise match would require estimating initial conditions, which is difficult. We added some sentences to clarify and highlight this. We hope that our revisiosn in section 4.4 in combination with our re-worked section 3.3 are effective and address your concerns.

14. *P25: "The feature-based approach reduces computational requirements only if we truly reduce the dimension of the data by focussing only on some of the features of the data." I think this is a useful comment and in some ways the focus on effective dimension is unfortunate because it obscures this component of the discussion. I wish a comment like this appeared more clearly back on P8.*

Thank you for pointing this out. We emphasized this point early on in our revised manuscript.

[revised manuscript text omitted]

---

## Author Response (AR2)

**Feature-based data assimilation in geophysics – response to reviewers (round two)**

Matthias Morzfeld, Jesse Adams, Spencer Lunderman, and Rafael Orozco

We thank the reviewers for patiently reading our paper and for pointing out a few more typos and one more minor issue (dimensionality of the prior/posterior distributions). All these issues have been corrected.